# InftyThink: Breaking the Length Limits of Long-Context Reasoning in Large Language Models

**Yuchen Yan**[1,2,*] **Yongliang Shen**[1,†] **Yang Liu**[2], **Jin Jiang**[2,3],
**Mengdi Zhang**[2], **Jian Shao**[1], **Yueting Zhuang**[1]
[1]Zhejiang University [2]Meituan Group [3]Peking University
{yanyuchen, syl}@zju.edu.cn

## Abstract

Advanced reasoning in large language models has achieved remarkable performance on challenging tasks, but the prevailing long-context reasoning paradigm faces critical limitations: quadratic computational scaling with sequence length, reasoning constrained by maximum context boundaries, and performance degradation beyond pre-training context windows. Existing approaches primarily compress reasoning chains without addressing the fundamental scaling problem. To overcome these challenges, we introduce InftyThink, a paradigm that transforms monolithic reasoning into an iterative process with intermediate summarization. By interleaving short reasoning segments with concise progress summaries, our approach enables unbounded reasoning depth while maintaining bounded computational costs. This creates a characteristic sawtooth memory pattern that significantly reduces computational complexity compared to traditional approaches. Furthermore, we develop a methodology for reconstructing long-context reasoning datasets into our iterative format, transforming OpenR1-Math into 333K training instances. Experiments across multiple model architectures demonstrate that our approach reduces computational costs while improving performance, with Qwen2.5-Math-7B showing 3-11% improvements across MATH500, AIME24, and GPQA_diamond benchmarks. Our work challenges the assumed trade-off between reasoning depth and computational efficiency, providing a more scalable approach to complex reasoning without architectural modifications.

**Project Page:** https://zju-real.github.io/InftyThink

## 1 Introduction

Recent studies have demonstrated the remarkable reasoning capabilities of large language models (LLMs), with models like OpenAI o1 (OpenAI, 2024), DeepSeek-R1 (Guo et al., 2025), Gemini 2.0 Flash Thinking (Google DeepMind, 2025), QwQ (Qwen Team, 2024), and Kimi-1.5 (Kimi Team et al., 2025) surpassing human performance on high-difficulty tasks including mathematical competitions (Latif et al., 2024b;a). These advanced reasoning models are typically developed through methodical techniques such as test-time scaling (Chen et al., 2024; Wan et al., 2024; Guan et al., 2025; Muennighoff et al., 2025), post-training on long-thought trajectories (Guo et al., 2025; Ye et al., 2025; Open Thoughts Team, 2025), or large-scale reinforcement learning (Guo et al., 2025; Shao et al., 2024) to generate effective reasoning paths that reach correct answers. A defining characteristic of these models is their ability to perform long-context reasoning, demonstrating advanced cognitive techniques including intent comprehension, multi-perspective analysis, self-reflection, and error correction (Zeng et al., 2024; Zhong et al., 2024). This evolution from simpler reasoning patterns to extensive deliberation has significantly improved problem-solving capabilities, particularly for complex challenges requiring multi-step inference.

---

[*]Contribution during internship at Meituan Group.
[†]Corresponding author.

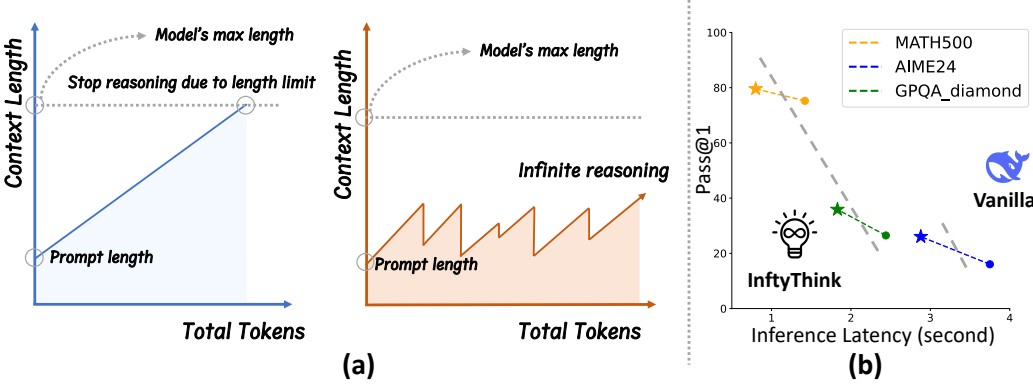

Figure 1: **(a)** Computational complexity comparison between vanilla long-context reasoning (blue, left) and InftyThink (orange, right). The sawtooth pattern of InftyThink demonstrates how periodic summarization creates a bounded memory footprint, substantially reducing computational costs (smaller area under curve) while enabling deeper reasoning. **(b)** InftyThink (⋆) reduces reasoning latency while simultaneously improving reasoning performance compared to the vanilla long-context reasoning (●).

However, this substantial improvement in reasoning quality comes with significant computational costs (OpenAI, 2024; Brown et al., 2024; Snell et al., 2024). The computational complexity of decoder-based LLMs grows quadratically with sequence length, resulting in prohibitive resource requirements for long-form reasoning. This efficiency bottleneck manifests in three primary challenges: **First**, current reasoning models often generate thousands of tokens even for moderately complex problems (OpenAI, 2024; Wang et al., 2025), creating substantial memory and processing overhead during inference. **Second**, reasoning processes are constrained by the model's maximum context length (aka. `max_length`) (Kuratov et al., 2024), frequently resulting in truncated reasoning that fails to reach conclusive answers (See analysis in Appendix E). **Third**, most LLM architectures are pre-trained with relatively small context windows (4k-8k tokens), causing performance degradation when reasoning extends beyond these boundaries (Li et al., 2024b; Yuan et al., 2025).

Existing approaches (Pang et al., 2024; Han et al., 2025; Zhang et al., 2025b) to address these limitations have explored various solutions with mixed success. Some methods attempt to compress reasoning chains post-generation (Xiao et al., 2023; Zhang et al., 2023; Chen et al., 2025), while others aim to train models to reason more concisely from the outset (Kang et al., 2024; Arora & Zanette, 2025; Liu et al., 2024; Xia et al., 2025). Chain-compression techniques like those employed in CoT-Valve (Ma et al., 2025) show promise but require predefined compression ratios during training, limiting their flexibility at inference time. TokenSkip (Xia et al., 2025) reduces redundant tokens by assessing each token's significance, though this impacts the model's reasoning performance. LightThinker (Zhang et al., 2025b) employs special tokens to dynamically compress the CoT process into a latent representation but lacks the ability to adaptively determine compression requirements for each step. Despite these advances, most approaches still operate within the traditional paradigm of generating a single, continuous reasoning chain, which merely attempting to make it more compact without addressing the fundamental computational scaling problem (A more detailed review of related work in Appendix C). This raises a critical question: ***Instead of optimizing within the constraints of monolithic reasoning, could we improve the model's accuracy and efficiency by altering its inherent reasoning paradigm?***

In this paper, we propose a fundamentally different approach to long-context reasoning. Rather than viewing reasoning as a single extended process, we introduce InftyThink, a novel paradigm that divides complex reasoning into multiple interrelated short reasoning segments. Each segment remains within a computationally efficient context length while maintaining the coherent flow of thought across iterations. This approach draws inspiration from human cognitive processes, where complex problem-solving frequently involves breaking problems into manageable parts and summarizing intermediate progress.

The core mechanism of InftyThink is an iterative process where the model generates a partial reasoning chain, summarizes its current thinking, and builds upon these summaries in subsequent iterations. As illustrated in Figure 1, traditional approaches (left, blue) face inevitable termination when context length reaches the model's maximum limit, often before completing the reasoning. In contrast, InftyThink (right, orange) creates a sawtooth pattern through periodic summarization, enabling unbounded reasoning depth while maintaining a bounded memory footprint. This approach both reduces computational complexity (smaller area under the curve) and overcomes the fundamental ceiling on reasoning depth imposed by context length constraints. Beyond computational efficiency, InftyThink offers a crucial advantage: it enables reasoning of arbitrary depth without architectural changes to the underlying model. By summarizing and building upon previous reasoning in manageable segments, models can effectively navigate complex problem spaces that would otherwise exceed context limitations.

To validate our approach, we reconstructed the existing SFT dataset OpenR1-Math, which was distilled from DeepSeek-R1, adapting it to conform to our proposed InftyThink paradigm. This reconstruction process transformed the original long-form reasoning examples into multiple interconnected reasoning segments with corresponding summaries. We then fine-tuned multiple base architectures on this reconstructed dataset and conducted comprehensive comparisons against traditional single-round long-context reasoning methods. Our experimental results demonstrate consistent improvements across various benchmarks, with Qwen2.5-Math-7B showing 2% improvement on MATH500, 11% improvement on AIME24, and 9% improvement on GPQA_diamond.

Our contributions are summarized as follows:

- We introduce InftyThink, which transforms monolithic long-form reasoning into iterative reasoning with summarization, mimicking human working memory patterns and reducing the quadratic computational complexity of transformer-based models to a more manageable form.

- We develop a technique to reconstruct existing long-context reasoning datasets (demonstrated on OpenR1-Math) into our iterative format, preserving reasoning quality while enabling more efficient computation without specialized architectures.

- Across multiple model architectures, our approach achieves significant improvements while substantially reducing computational costs, challenging the assumed trade-off between reasoning depth and efficiency.

## 2 METHODS

In this section, we present InftyThink, a novel reasoning paradigm that addresses the context limit and computational inefficiency of vanilla long-context reasoning in large reasoning models. First, we formalize our proposed iterative reasoning framework that enables unbounded reasoning depth while maintaining a bounded memory footprint (Section 2.1). Then, we detail a principled approach for reconstructing existing long-context reasoning datasets to conform to our paradigm (Section 2.2).

### 2.1 INFTYTHINK REASONING PARADIGM

To address the computational challenges inherent in long-context reasoning, we propose InftyThink, a novel paradigm that re-imagines how LLMs approach complex reasoning tasks. This paradigm decomposes complex reasoning into a series of bounded-length segments with intermediate summarization steps, enabling theoretically unlimited reasoning depth without the quadratic computational scaling of traditional approaches. Figure 2 illustrates the key differences between our approach and vanilla reasoning. Below, we first formalize the conventional reasoning approach before presenting our iterative framework.

### 2.1.1 VANILLA PARADIGM OF LONG-CONTEXT REASONING

Contemporary reasoning models, particularly those in the class of DeepSeek-R1 and similar architectures, rely on extended single-round generation for complex reasoning tasks. These models generate content comprising two principal components: a comprehensive "thinking" phase that captures the exploratory reasoning process, followed by a "conclusion" phase that distills key insights into a

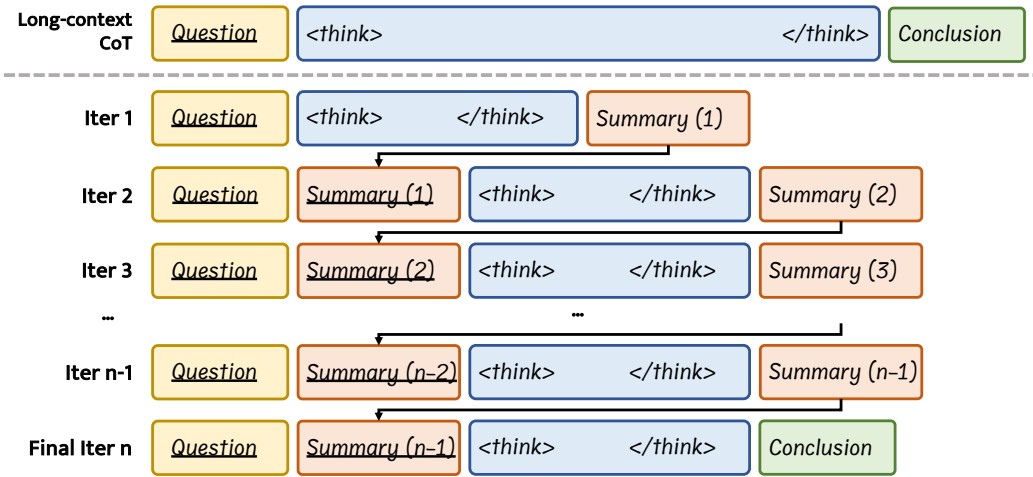

Figure 2: Illustration of InftyThink versus vanilla long-context reasoning. **Upper panel:** Vanilla long-context reasoning generates continuous tokens until reaching maximum context length. **Lower panel:** Our InftyThink approach divides reasoning into multiple iterations. The underlined segments represent content included in the prompt as model input, while non-underlined segments show model-generated output. Each iteration in InftyThink consists of: (1) summarizing previous reasoning progress, (2) generating a focused reasoning segment within an efficient token budget, and (3) producing a concise progress summary. This iterative process enables arbitrarily deep reasoning chains without architectural modifications to the underlying model, while maintaining significantly lower computational complexity compared to traditional approaches.

structured response. This conventional reasoning paradigm can be formalized as:

$$\texttt{<|U|>}Q\texttt{<|A|>}\underline{\texttt{<think>}RP\texttt{</think>}C}$$

where `<|U|>` and `<|A|>` are usually special tokens defined in the chat template to mark the dialogue structure, $Q$ denotes the user query, `<think>` and `</think>` encapsulate the model's reasoning process $RP$, and the final conclusion $C$ synthesizes the reasoning into a coherent answer. The underlined part represents the model's output, while the rest serves as the prompt input to the model.

This established approach, while effective for many problems, faces a fundamental limitation: as reasoning complexity increases, the token length of $RP$ grows substantially, often exceeding context window constraints and incurring quadratic computational costs. To address this limitation, we introduce InftyThink, a paradigm that transforms monolithic reasoning into an iterative process with intermediate summarization steps.

### 2.1.2 ITERATIVE REASONING WITH SUMMARIZATION: INFTYTHINK

In the InftyThink paradigm, reasoning proceeds through multiple connected segments, each maintaining computational efficiency while preserving the coherent progression of thought. The initial reasoning iteration is formalized as:

$$\texttt{<|U|>}Q\texttt{<|A|>}\underline{\texttt{<think>}RP_1\texttt{</think><summary>}S_1\texttt{<summary>}}$$

where $RP_1$ represents the first segment of reasoning constrained to an efficient length, and $S_1$ denotes a concise summary of this segment encapsulated by the special tokens `<summary>` and `</summary>`. This summary serves as a compressed representation of the reasoning state, capturing essential information while discarding unnecessary details.

For subsequent iterations ($i > 1$), the model builds upon previous reasoning by incorporating the prior summary:

$$\texttt{<|U|>}Q\texttt{<|A|>}\underline{\texttt{<history>}S_{i-1}\texttt{</history>}\texttt{<think>}RP_i\texttt{</think><summary>}S_i\texttt{</summary>}}$$

where `<history>` and `</history>` delimit the previous summary $S_{i-1}$, which provides critical context for the current reasoning segment $RP_i$. Each iteration maintains a bounded token length while building upon accumulated knowledge through the summary mechanism.

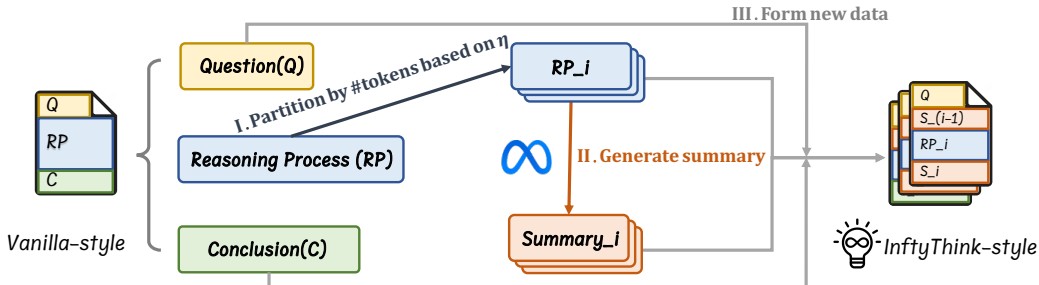

Figure 3: Systematic pipeline for reconstructing vanilla-style long-context reasoning data into the InftyThink-style format. **I.** Original reasoning processes are partitioned into optimally sized fragments based on parameter ($\eta$), preserving semantic coherence. **II.** Meta-Llama-3.3-70B-Instruct generates concise yet comprehensive summaries for each reasoning fragment. **III.** The original fragments and their generated summaries are systematically recombined to create InftyThink-style training instances that teach the model to reason iteratively.

The final iteration ($n$) culminates in a conclusion rather than another summary:

$$\texttt{<|U|>}Q\texttt{<|A|>}\texttt{<history>}S_{n-1}\texttt{</history>}\texttt{<think>}RP_n\texttt{</think>}C$$

Throughout these expressions, blue denotes reasoning segments, orange represents intermediate summaries, and green indicates the final conclusion. This formulation elegantly handles edge cases: when problems are simple enough to be solved in a single iteration, the model bypasses summary generation, defaulting to the vanilla paradigm.

During inference, the model iteratively generates reasoning segments and corresponding summaries, with each summary becoming the context for the subsequent iteration. This process continues until the model produces a conclusion instead of a summary, signaling completion of the reasoning task. To prevent potential infinite loops, we impose a hyperparameter `max_iters` that terminates iteration if exceeded, though our empirical results indicate that well-trained models naturally converge within a reasonable number of iterations.

## 2.2 DATA RECONSTRUCTION

While our InftyThink paradigm offers a theoretically compelling approach to unbounded reasoning, it requires appropriate training data to enable models to learn this iterative reasoning process. Prior work has established that models can acquire sophisticated reasoning capabilities through supervised fine-tuning on data generated by highly capable reasoners (Guo et al., 2025; open-r1, 2025). Building on this insight, we develop a principled methodology for transforming existing long-context reasoning datasets into our iterative format.

We select OpenR1-Math (open-r1, 2025)[1] as our source dataset, which is a collection of mathematical reasoning generated by DeepSeek-R1 in response to questions from NuminaMath-1.5 (Jia LI et al., 2025). This dataset spans a diverse spectrum of mathematical domains and difficulty levels, from elementary mathematics to competition-level problems, making it an ideal testbed for our approach.

In addition, we also performed similar data reconstruction and related experiments on the OpenThoughts (Guha et al., 2025) dataset to demonstrate the robustness of our method. Detailed results are provided in Appendix M. We also report the resource consumption of the data construction process in Appendix G, enabling researchers to assess the feasibility of adopting our method.. Our reconstruction pipeline comprises three key stages:

**Step I: Reasoning Process Partition** For each instance in the dataset, we partition the original reasoning process ($RP$) into segments based on a hyperparameter $\eta$ that determines the maximum token length of each segment. Rather than applying arbitrary truncation, we implement a semantically-aware segmentation algorithm: we first decompose the reasoning process into semantic

---

[1]All data usage in this paper is in full compliance with the terms and conditions of the Apache License 2.0.

units by identifying natural breakpoints at sentence or paragraph boundaries. These units are then tokenized and sequentially aggregated into segments, optimizing for coherence while ensuring each segment remains below the $\eta$ threshold. This process yields a sequence of reasoning segments $RP_1, RP_2, \ldots, RP_n$, formally expressed as:

$$\text{Partition}(RP, \eta) \Rightarrow RP_1, RP_2, \ldots, RP_n \tag{1}$$

**Step II. Summary Generation** For each reasoning segment, we generate a concise summary that captures its essential insights and progress toward the solution. To ensure information continuity across iterations, we employ a sophisticated foundation model $M$ (specifically Meta-Llama-3.3-70B-Instruct (Grattafiori et al., 2024), and in Appendix F, we provide a performance comparison of summary generation using different foundation models. The experimental results demonstrate that the size of the summarizer model has no significant impact on the performance of InftyThink.) with carefully crafted prompting (provided in Appendix P):

$$S_i = \text{summarize}(M, RP_i, \{RP_1, \ldots, RP_{i-1}\}) \tag{2}$$

The summarization model receives not only the current reasoning segment $RP_i$ but also all preceding segments and their summaries, enabling it to create summaries that maintain reasoning continuity. In the Appendix I, we further examine the necessity of preserving a global summary throughout the reasoning process. Our analysis substantiates the design choice of constructing $S_i$ from the cumulative reasoning trajectory $(RP_1, \ldots, RP_i)$, confirming that this formulation is both reasonable and essential for maintaining global coherence in the InftyThink paradigm.

**Step III. Training Instance Construction** From the segmented reasoning and generated summaries, we construct training instances that teach the model to perform iterative reasoning with summarization. These instances follow the structure of our InftyThink paradigm:

$$D_i = \begin{cases} (Q, RP_1, S_1) & \text{for } i = 1, \\ (Q, S_{i-1}, RP_i, S_i) & \text{for } 1 < i < n, \\ (Q, S_{n-1}, RP_n, C) & \text{for } i = n. \end{cases} \tag{3}$$

For the initial reasoning step ($i = 1$), the model learns to generate the first reasoning segment followed by its summary. For intermediate steps ($1 < i < n$), it learns to continue reasoning based on previous summaries and generate new summaries. For the final step ($i = n$), it learns to produce a conclusive answer. This reconstruction process transforms each original example into $n$ training instances, where $n$ is the number of reasoning segments. The complete pipeline is illustrated in Figure 3. Applying this methodology to the OpenR1-Math dataset with $\eta$=4k, we expand the original 220K examples into 333K training instances, forming our InftyThink-style OpenR1-Math-Inf dataset. This dataset enables models to learn the InftyThink approach through supervised fine-tuning. Since our method generates additional data samples, we provide a detailed discussion in Appendix H and demonstrate that such a comparison is both fair and reasonable.

## 3 EXPERIMENTS

### 3.1 SETTINGS

We employ instruction fine-tuning to validate the proposed reasoning paradigm and associated dataset. Specifically, akin to the distilled model discussed in DeepSeek-R1 (Guo et al., 2025), training is conducted on five base models of varying sizes: Qwen2.5-Math-1.5B, Qwen2.5-Math-7B (Yang et al., 2024), Qwen2.5-14B, Qwen2.5-32B (Qwen et al., 2025), and Meta-Llama-3.1-8B (Grattafiori et al., 2024), and two instruct models: Qwen2.5-Math-1.5B-Instruct and DeepSeek-R1-Distill-Qwen-1.5B (Guo et al., 2025). Instruction-based fine-tuning is applied using both OpenR1-Math (for vanilla) and the newly introduced OpenR1-Math-Inf (for InftyThink). The hyperparameters are configured with $\eta$ set to 4k and `max_iters` set to 10. The detailed experimental setup is provided in the Appendix O. The trained models are evaluated across multiple benchmarks, including MATH500 (Hendrycks et al., 2021; Lightman et al., 2023), AIME24, and GPQA_diamond (Rein et al., 2024). We also provide evaluations on more challenging benchmarks in Appendix K, including AIME25, Math Odyssey (Fang et al., 2025), and AMC23 (mathai, 2023). To demonstrate the broad applicability, we also conducted evaluations on code tasks like HumanEval (Chen et al., 2021) and MBPP (Austin et al., 2021), presented in Appendix M and N. In addition, we conducted a stability analysis of the training and evaluation process, and the results are presented in Appendix J.

Table 1: Our main experimental results. The results are obtained by sampling the model 16 times with a temperature of 0.7. **ACC** stands for average accuracy(%), **TOK** stands for average number of generated tokens (K), and **LAT** stands for average inference wall time in seconds.

| Model | Train Format | MATH500 | | | AIME24 | | | GPQA_diamond | | | Average | | |
|---|---|---|---|---|---|---|---|---|---|---|---|---|---|
| | | ACC↑ | TOK | LAT↓ | ACC↑ | TOK | LAT↓ | ACC↑ | TOK | LAT↓ | ACC↑ | TOK | LAT↓ |
| *Base Models* | | | | | | | | | | | | | |
| Qwen2.5-Math-1.5B | Vanilla | 75.24 | 5.94 | 1.42 | 16.04 | 16.03 | 3.75 | 26.48 | 10.51 | 2.44 | 59.54 | 7.60 | 1.79 |
| | InftyThink | 79.57 | 6.79 | 0.80 | 26.04 | 20.16 | 2.88 | 35.89 | 10.59 | 1.83 | 65.48 | 8.38 | 1.17 |
| Qwen2.5-Math-7B | Vanilla | 89.51 | 4.32 | 1.26 | 32.92 | 14.26 | 4.15 | 43.94 | 8.95 | 2.39 | 74.78 | 5.99 | 1.69 |
| | InftyThink | 91.29 | 4.72 | 0.76 | 43.96 | 22.11 | 4.66 | 52.97 | 9.54 | 2.63 | 78.92 | 6.75 | 1.43 |
| Llama-3.1-8B | Vanilla | 82.10 | 5.50 | 2.58 | 20.83 | 16.70 | 10.01 | 41.35 | 10.33 | 5.02 | 68.49 | 7.27 | 3.55 |
| | InftyThink | 82.28 | 6.28 | 2.35 | 34.17 | 20.87 | 12.99 | 47.51 | 10.30 | 4.81 | 70.84 | 7.97 | 3.46 |
| Qwen2.5-14B | Vanilla | 93.86 | 3.88 | 1.49 | 48.75 | 13.87 | 11.30 | 57.42 | 7.71 | 3.24 | 82.09 | 5.33 | 2.37 |
| | InftyThink | 93.07 | 3.38 | 1.43 | 51.67 | 18.66 | 7.11 | 59.44 | 9.04 | 3.08 | 82.22 | 5.55 | 2.11 |
| Qwen2.5-32B | Vanilla | 96.24 | 2.51 | 1.91 | 57.11 | 11.96 | 13.43 | 63.45 | 7.07 | 4.31 | 85.71 | 4.14 | 3.04 |
| | InftyThink | 95.75 | 3.84 | 1.48 | 64.38 | 16.18 | 8.88 | 68.18 | 7.86 | 4.14 | 86.93 | 4.99 | 2.58 |
| *Instruct Models* | | | | | | | | | | | | | |
| Qwen2.5-Math-1.5B | / | 75.11 | 0.55 | 0.03 | 8.54 | 0.97 | 0.08 | 25.32 | 1.19 | 0.51 | 58.82 | 0.74 | 0.16 |
| (-Instruct) | Vanilla | 73.99 | 3.89 | 1.64 | 11.67 | 17.49 | 5.16 | 17.77 | 12.42 | 3.51 | 56.13 | 6.77 | 2.29 |
| | InftyThink | 81.10 | 6.51 | 0.87 | 26.04 | 20.05 | 3.20 | 27.87 | 12.10 | 2.59 | 64.35 | 8.59 | 1.43 |
| R1-distill-Qwen-1.5B | / | 85.00 | 5.25 | 1.10 | 22.50 | 17.92 | 5.41 | 37.18 | 8.29 | 1.59 | 69.42 | 6.59 | 1.41 |
| | Vanilla | 85.54 | 5.07 | 1.09 | 24.79 | 14.03 | 3.18 | 34.69 | 10.01 | 2.32 | 69.21 | 6.78 | 1.51 |
| | InftyThink | 88.06 | 5.75 | 0.89 | 29.38 | 21.37 | 4.96 | 35.54 | 12.80 | 1.21 | 71.36 | 8.31 | 1.14 |

## 3.2 MAIN RESULTS

Table 1 presents our comprehensive evaluation of InftyThink across five base model architectures and two instruct model architectures of varying scales and specializations. We provide several actual reasoning trajectories of vanilla CoT and InftyThink style on MATH500, AIME24, and GPQA_diamond in the Appendix U to facilitate a clearer understanding of how InftyThink works. Several important patterns emerge from these results that provide insight into how our proposed reasoning paradigm affects model performance.

**Consistent Improvements Across Model Families and Scales.** Our InftyThink consistently outperforms the vanilla reasoning approach across all model sizes and architectures. Notably, the improvements generalize beyond the Qwen architecture family to Meta-Llama-3.1-8B, demonstrating that the benefits of our iterative reasoning paradigm are not architecture-specific but rather represent a fundamental improvement in how models approach complex reasoning.

**Extended Reasoning Depth and Decreased Inference Latency.** InftyThink mitigates the computational overhead associated with the $O(n^2)$ complexity of LLMs at extended inference lengths by decomposing a single long generation into multiple shorter generation steps. This approach consistently enhances throughput across LLMs of varying parameter scales. Furthermore, the iterative mechanism inherent in InftyThink allows models to efficiently handle extended reasoning tasks, maintaining high inference speed even as the generation length significantly increases. We further discuss this aspect in Appendix R.

**Scaling Trends with Model Size.** We observe an interesting relationship between model scale and the magnitude of improvement from InftyThink. The relative gains are most pronounced in smaller models (e.g., 4.33%, 10.00%, and 9.41% improvements for Qwen2.5-Math-1.5B on the three benchmarks) and gradually diminish as model size increases, particularly on the MATH500 benchmark. This suggests that InftyThink provides a form of algorithmic enhancement that partially compensates for limited model capacity, effectively allowing smaller models to perform more complex reasoning than their size would typically permit.

The iterative summarization mechanism in InftyThink appears to effectively mitigate the limitations of traditional long-context reasoning by enabling more structured exploration of the solution space. The pattern of improvements suggests that our approach particularly benefits complex problems

requiring multi-step reasoning, which are precisely the scenarios where long-context reasoning is most challenged by computational constraints. Our findings also suggest important implications for model scaling: InftyThink may offer a more computationally efficient path to improved reasoning capabilities than simply scaling model size, particularly for smaller models where the relative improvements are most pronounced. This could have significant practical implications for deploying advanced reasoning capabilities in resource-constrained environments.

## 4 ANALYSIS

### 4.1 ENDOWING SHORT-CONTEXT MODELS WITH LONG-CONTEXT REASONING ABILITY

Many foundational LLMs are pretrained with limited context windows (4k or 8k tokens), yet a significant portion of reasoning datasets exceeds these boundaries. Analysis of OpenR1-Math shows only 54% of samples contain fewer than 4k tokens, and 83% are within 8k tokens (Appendix D), revealing a critical mismatch between model architecture and reasoning requirements.

Table 1 shows consistent performance improvements when applying InftyThink across all model configurations. The gains are particularly notable on complex benchmarks like AIME24 and GPQA_diamond, where problems typically require longer reasoning chains that would exceed standard context windows. For instance, Qwen2.5-Math-7B achieves a 11.04% improvement on AIME24 and a 9.03% improvement on GPQA_diamond using our approach. These improvements suggest that InftyThink effectively addresses context length limitations by restructuring long reasoning into manageable segments with summarization.

To validate our approach against alternative context extension methods, we implemented RoPE positional encoding interpolation (Su et al., 2023; Chen et al., 2023; kaiokendev, 2023), a common technique for extending context windows beyond pretraining lengths. While this approach yielded modest improvements, it consistently underperformed compared to InftyThink across all benchmarks (detailed results in Appendix Q). This comparison is particularly revealing: rather than attempting to stretch architectural limitations through embedding manipulation, InftyThink restructures the reasoning process itself to work within existing constraints.

These findings suggest that InftyThink offers a more effective solution to the long-context reasoning challenge than traditional context window extension techniques. By allowing models to periodically summarize and build upon previous reasoning, our approach enables more flexible and adaptable reasoning capabilities that aren't bound by fixed architectural constraints. This has important implications for deploying reasoning systems in environments where context length would otherwise be a limiting factor.

### 4.2 INFLUENCE OF CONTEXT WINDOW SIZE PARAMETER $\eta$

Parameter $\eta$ plays a crucial role in InftyThink, controlling the maximum token length for each reasoning iteration. This parameter creates a fundamental tradeoff: larger values reduce the number of iterations but increase per-iteration computational cost, while smaller values distribute computation more evenly but potentially fragment reasoning.

Table 2 presents performance across three $\eta$ values (2k, 4k, and 6k) on benchmarks, with all configurations consistently outperform the baseline with no clear optimal value across all datasets. On MATH500, performance increases marginally with $\eta$, suggesting that longer uninterrupted reasoning benefits simpler problems.

These results challenge the intuition that fragmented reasoning necessarily harms performance. Even with $\eta$=2k, where reasoning is interrupted every 2k tokens, InftyThink main-

Table 2: Evaluation results across different $\eta$. GPQA_D refers to GPQA_diamond. Experiments are conducted on Qwen2.5-Math-7B.

| Method | $\eta$ | MATH500 | AIME24 | GPQA_D |
|---|---|---|---|---|
| Vanilla | / | 89.51 | 32.92 | 43.94 |
| InftyThink | 2k | 90.84+1.33 | 38.29+5.37 | 48.96+5.02 |
| | 4k | 91.29+1.78 | 43.96+11.04 | 52.97+9.03 |
| | 6k | 91.16+1.65 | 39.29+6.37 | 49.87+5.93 |

tains or improves performance across all benchmarks. This suggests that well-designed summarization mechanisms effectively preserve critical information while discarding redundant computation.

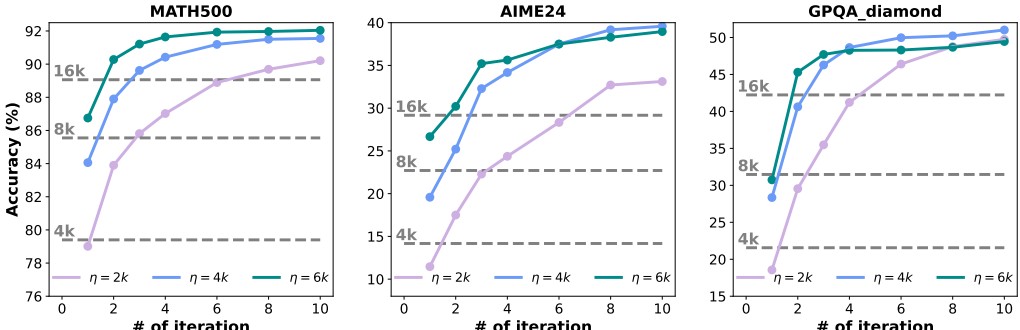

Figure 4: Model performance across iterations compared to vanilla reasoning. Horizontal lines represent vanilla long-context reasoning with different `max_new_tokens` settings (4k, 8k, 16k). Curves show InftyThink's accuracy evolution across iterations under different $\eta$ settings (2k, 4k, 6k). The data points in the figure represent the accuracy of the model under each corresponding setting when allowed up to $n$ iterations, including those reasoning trajectories that finished in fewer than $n$ iterations. InftyThink rapidly surpasses fixed-length reasoning constraints, with performance continuing to improve beyond traditional token limits. Experiments conducted on Qwen2.5-Math-7B.

The robustness to different $\eta$ values demonstrates that InftyThink's benefits derive primarily from its iterative summarization approach rather than specific segmentation boundaries. This flexibility allows practitioners to select $\eta$ values based on hardware constraints or specific application requirements without significant performance penalties. A larger $\eta$ reduces the number of summarization steps and better leverages existing GPU optimizations, while a smaller $\eta$ reduces the number of generated tokens per iteration, helping maintain a lower GPU memory footprint.

## 4.3 PERFORMANCE ACROSS REASONING ITERATION ROUNDS

A defining characteristic of InftyThink is its ability to transcend the constraints of maximum context length through iterative reasoning with summarization. Figure 4 quantifies this capability by tracking performance across reasoning iterations and comparing it against traditional reasoning methods with fixed token limits. In Appendix L, we report the model's performance dynamics on AIME25, AMC23, and Math Odyssey across inference iterations. The results reveal three key insights:

**Progressive performance improvement.** Unlike traditional reasoning approaches that hit a performance ceiling determined by their maximum token limit, InftyThink enables continuous improvement through successive iterations. On AIME24, a challenging benchmark, performance steadily increases from iterations 1 through 10 for all $\eta$ settings, demonstrating that complex problems benefit substantially from extended reasoning beyond conventional context limits.

**Efficiency of iterative summarization.** Even with the smaller context setting of $\eta$=2k, InftyThink eventually reaches comparable performance to larger $\eta$ values across all benchmarks. This is particularly evident on AIME24, where by iteration 10, the $\eta$=2k configuration approaches the performance of $\eta$=6k despite using reasoning segments one-third the size. This demonstrates that effective summarization can preserve critical reasoning information even with frequent compression. At different values of $\eta$, as $n$ increases, the model can gradually activate its internal reasoning capabilities and effectively solve the problem. We observe that models with $\eta$=2k require more iterations to reach the performance level of those with $\eta$=4k or $\eta$=6k.

**Early-stage performance tradeoffs.** Models with larger $\eta$ values (6k) consistently outperform those with smaller segments (2k) in early iterations across all benchmarks. However, this advantage diminishes and sometimes reverses in later iterations, particularly on GPQA_diamond where $\eta$=4k eventually surpasses $\eta$=6k. This suggests that while larger segments provide initial advantages, they may commit the model to reasoning paths that become difficult to revise, whereas smaller segments allow more flexible exploration over multiple iterations.

### 4.4 ABLATION ON PROMPT-BASED PIPELINE

Our work introduces a data construction pipeline that converts vanilla CoT–formatted reasoning traces into the InftyThink format. Fine-tuning LLMs on these reconstructed datasets enables a transition from the vanilla CoT paradigm to the iterative InftyThink reasoning paradigm. A central question is the extent to which training contributes to this transition,specifically, whether a purely prompt-based approach could suffice. That is, *can we summarize intermediate reasoning within a fixed window, propagate these summaries forward, and induce iterative reasoning without any additional training?* To investigate this, we design an ablation experiment to evaluate the necessity of both our data reconstruction method and the fine-tuning process.

Concretely, we take DeepSeek-R1-distill-1.5B as the base reasoner model and let it perform standard vanilla-style reasoning. Once the generated reasoning reaches a predefined length $\eta$, we truncate its output and feed the completed portion into a summarizer model to produce a summary. We then prompt the reasoner to continue reasoning conditioned on this summary, iteratively generating new summaries and continuing the reasoning until completion. In our setup, the hyperparameter is fixed at $\eta$=4k. We additionally evaluate Qwen2.5-1.5B-Instruct and Llama3.3-70B-Instruct as summarizers. The full experimental results are presented in Table 3.

Table 3: Ablation results (%) of training-free prompt-based iterative reasoning.

| Reasoner model | Summarizer model | MATH500 | AIME24 | GPQA_D |
|---|---|---|---|---|
| *Training-based approaches* | | | | |
| R1-distill-Qwen-1.5B | / | 85.54 | 24.79 | 34.69 |
| (tuned on OpenR1-Math) | InftyThink | 88.06 | 29.38 | 35.54 |
| *Training-free approaches* | | | | |
| R1-distill-Qwen-1.5B | R1-distill-Qwen-1.5B | 83.63 | 26.85 | 26.76 |
| (tuned on OpenR1-Math) | Qwen2.5-1.5B-Instruct | 82.97 | 20.45 | 30.92 |
| | Llama3.3-70B-Instruct | 84.14 | 26.65 | 31.88 |

From our experimental results, directly applying a prompt-based method to implement iterative reasoning leads to a slight performance drop. The primary reason is that this inference paradigm is not extensively trained and thus deviates from the patterns the model was exposed to during training. As a result, the model may lose important information across multiple reasoning iterations, preventing it from fully leveraging its intrinsic reasoning capability.

In addition, we observe that implementing InftyThink via a prompt-based approach introduces significantly higher latency compared to a training-based approach. Each iteration requires extracting the current reasoning trace and generating an additional summary, which increases the overall computational load during inference and undermines the efficiency gains that InftyThink is designed to provide in reducing inference wall-time.

## 5 CONCLUSION

In this paper, we introduced InftyThink, a novel reasoning paradigm that transforms monolithic long-context reasoning into an iterative process with periodic summarization. By generating partial reasoning, summarizing current understanding, and building upon these summaries in subsequent iterations, our approach effectively addresses the quadratic computational complexity and context length limitations of conventional approaches. Experiments across multiple model architectures demonstrate consistent performance and throughput improvements on challenging reasoning benchmarks . Our analysis confirms that InftyThink not only reduces computational costs but also enables models to transcend their native context window limitations without architectural modifications. We further discuss the limitations of InftyThink and its future directions in Appendix B. InftyThink paradigm represents a step toward more cognitively plausible AI reasoning through iterative refinement rather than exhaustive single-pass analysis, opening promising avenues for more efficient, flexible reasoning in language models that decouples reasoning depth from computational complexity.

ACKNOWLEDGEMENT

This work was supported by National Natural Science Foundation of China (No. 62506332).

ETHICS STATEMENT

This work does not involve human subjects, personal data, or sensitive information. All datasets used in our experiments are publicly available datasets designed for evaluating mathematical reasoning in LLMs. We strictly adhered to ethical research practices and did not conduct any data collection that could raise privacy, security, or fairness concerns. Our methods do not introduce risks of harmful applications. To the best of our knowledge, this research complies with the ICLR Code of Ethics and poses no foreseeable ethical concerns.

REPRODUCIBILITY STATEMENT

We provide the implementation of InftyThink in the supplementary materials, including scripts for data paradigm conversion and inference code, to facilitate the adoption of our method by researchers. In addition, we will release an open-source version of InftyThink-style data.

Regarding experiments and evaluations, we include detailed descriptions of our training and evaluation procedures in Appendix O to enable researchers to reproduce our work.

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

CONTENTS

## A    LLM Usage Declaration

In writing this paper, we only used LLMs for polishing. The generation of ideas in this work **did not** involve any assistance from LLMs. The experimental design and manuscript writing were **not directly produced by LLMs** either. The models were used solely as a polishing tool: specifically, we first drafted the manuscript, then refined it with the help of an LLM, and finally the authors conducted another round of verification after polishing.

## B    Discussion

**Alignment with Human Reasoning**    InftyThink's iterative reasoning approach shares interesting parallels with human problem-solving strategies. Humans rarely solve complex problems through a single, exhaustive thought process but instead work through incremental steps, summarizing intermediate progress, and building on previous insights. The strong performance of our approach, particularly on complex problems, suggests that structuring AI reasoning to better align with these natural problem-solving patterns may yield both efficiency and effectiveness benefits. This connection between iterative reasoning with summarization and improved performance offers potential insights for developing more effective AI reasoning systems.

**Adaptive Reasoning Depth**    Unlike conventional reasoning approaches with fixed computational budgets, InftyThink can adaptively allocate computational resources based on problem difficulty. Our analysis in Section 4.3 shows that simpler problems (e.g., in MATH500) reach ceiling performance with fewer iterations, while more complex problems (in AIME24 and GPQA_diamond) benefit from extended reasoning. This adaptive depth capability has important implications for practical deployment, as it enables efficient resource allocation across heterogeneous problem sets without requiring predetermined computation limits.

**Limitations**    Despite its advantages, InftyThink faces several limitations. First, the quality of reasoning depends heavily on the model's summarization capabilities—poor summarization can lead to information loss that hinders subsequent reasoning. Second, breaking reasoning into segments might disrupt the coherent flow of thought for certain problem types that benefit from maintaining a complete chain of reasoning. Finally, while our approach reduces computational complexity, it introduces additional inference steps that may increase latency in time-sensitive applications.

**Future Directions**    Several promising directions could extend this work. First, integrating reinforcement learning techniques such as GRPO could help models better learn when and what to summarize, potentially improving information retention across iterations (Yan et al., 2026). Second, exploring variable-length reasoning segments that adapt based on problem complexity could further optimize the tradeoff between computational efficiency and reasoning coherence. Third, applying InftyThink to multi-modal reasoning tasks could expand its applicability to domains requiring integration of visual, textual, and numerical reasoning. Finally, investigating how to parallelize different reasoning paths within the InftyThink framework could further accelerate complex problem-solving.

## C    Related Works

### C.1    Reasoning of Large Language Models

Reasoning ability is one of the fundamental competencies of large language models(LLMs), reflecting their capacity to tackle complex challenges in the human domain. Currently, LLMs demonstrate impressive performance across various reasoning tasks, including commonsense reasoning, mathematical reasoning, code reasoning, logical reasoning and etc. The reasoning capabilities of these models can be enhanced at several stages during training (Huang & Chang, 2023; Sun et al., 2024). For instance, during pre-training, the inclusion of extensive reasoning-related knowledge and examples helps the model to learn reasoning patterns from the data (Shao et al., 2024; Guo et al., 2024; Rozière et al., 2024). Similarly, in the supervised fine-tuning phase, incorporating high-quality reasoning question-answer pairs can further refine the model's reasoning patterns and enhance its capabilities (Li et al., 2024a; Yan et al., 2025; Yu et al., 2023; Jiang et al., 2025). In the reinforcement learning

phase, the model's reasoning is monitored and guided through feedback on outcome or processes, providing additional improvements (Guo et al., 2025; Shao et al., 2024). The release of OpenAI's o1 (OpenAI, 2024) marked a significant breakthrough in the reasoning abilities of LLMs. OpenAI o1 demonstrated long-context reasoning capabilities, where the model utilized extended chains of thought to integrate planning, self-correction, and other cognitive functions, significantly boosting its reasoning performance (Zeng et al., 2024). More recently, DeepSeek-R1, an open-source o1-like reasoning model, has exhibited comparable reasoning abilities. Furthermore, distilled data from DeepSeek-R1 enables smaller LLMs to also acquire long-context reasoning skills (Guo et al., 2025; Ye et al., 2025; Open Thoughts Team, 2025).

### C.2 COMPRESSION OF LLM'S REASONING PROCESS

Current research on compressing the Chain-of-Thought (CoT) process to accelerate large language model (LLM) inference is primarily categorized into two approaches: CoT token compression and KV cache compression. CoT token compression enhances inference efficiency by reducing the number of tokens generated by the model. This approach can be further subdivided into discrete text token compression and continuous latent token compression methods. Discrete text token compression employs straightforward strategies such as prompt engineering (Han et al., 2025), instruction fine-tuning (Kang et al., 2024), and reinforcement learning (Arora & Zanette, 2025) to train models to produce more concise reasoning processes. Within this category, the skip-tokens method (Liu et al., 2024; Xia et al., 2025) enables the model to intelligently skip unimportant tokens during inference, thereby achieving acceleration. In contrast, continuous latent token compression (Cheng & Durme, 2024; Geiping et al., 2025; Hao et al., 2025; Shen et al., 2025) explores a more innovative approach by attempting to compress reasoning steps into continuous latent representations. This allows LLMs to perform effective inference without explicitly generating discrete word tokens. On the other hand, KV cache compression optimizes inference performance by reducing the storage requirements and computational load of the KV cache. This approach mainly includes two types of methods: training-free and training-based. Training-free KV cache management strategies enhance efficiency by selectively retaining key tokens. The criteria for selection include prioritizing initial and most recent tokens for their temporal relevance (Xiao et al., 2023), identifying tokens with significant historical attention (Zhang et al., 2023), or selecting tokens based on structural cues such as punctuation marks (Chen et al., 2025). Training-based KV cache management (Pang et al., 2024; Zhang et al., 2025b) involves introducing special tokens and training LLMs to compress important historical information into these tokens, thereby achieving KV cache merging. This method instructs the model on when to perform compression during the training phase and applies corresponding interventions during the inference phase. With the application of reinforcement learning (RL) in large language models (LLMs), researchers have also employed RL to achieve efficient reasoning. A typical approach is to incorporate penalties on the length of model generations into the reward design (Aggarwal & Welleck, 2025; Arora & Zanette, 2025; Kimi Team et al., 2025; Luo et al., 2025; Lyu et al., 2025; Wu et al., 2025; Zhang et al., 2025a). By dynamically penalizing the length of the model's reasoning, the overall reasoning length can be effectively reduced. However, this also suppresses the model's exploration space when tackling difficult problems.

### C.3 ITERATIVE METHODS IN LLMS

Large language models (LLMs) often face the problem of context window length limitations in generation tasks. For example, in novel generation, when the model produces millions of tokens, such issues frequently arise. This is because LLMs are typically trained with a fixed context window, and once the generation exceeds this limit, performance degradation often occurs. An intuitive approach is to periodically summarize the content generated by the model, condensing ultra-long text while preserving the core semantic information, thereby reducing token usage. RecurrentGPT (Zhou et al., 2023) is a prompting-based iterative generation method that maintains both long-term and short-term memory, continuously updating the model's current task in an attempt to address the issue of limited context window in long-form generation tasks. Tree-of-Thoughts (Yao et al., 2023) decomposes a problem into multiple subproblems, performs verification at each subproblem, and selects the optimal path to obtain a more reliable reasoning trajectory, which could be used as a test-time scaling method that improves performance through exploration and verification ReAct (Yao et al., 2022) explores the domain of tool-use by integrating reasoning into the model's action space to enhance its tool-calling capabilities, and during tool usage, the model needs to iteratively plan, summarize, reason,

and generate. Least-to-most (Zhou et al., 2022) solves simpler subproblems first to provide useful context for solving more complex ones later. Through an iterative generation process, the model can effectively leverage its contextual information to perform efficient reasoning and generation.

## D    TOKEN LENGTH DISTRIBUTION OF OPENR1-MATH

To analyze the characteristics of the OpenR1-Math dataset, we tokenized all data using the Qwen2.5-Math-7B tokenizer and computed the distribution. As shown in Figure 5, 54% of the samples in OpenR1-Math are shorter than 4k tokens, 83% are shorter than 8k tokens, and the vast majority fall below 16k tokens.

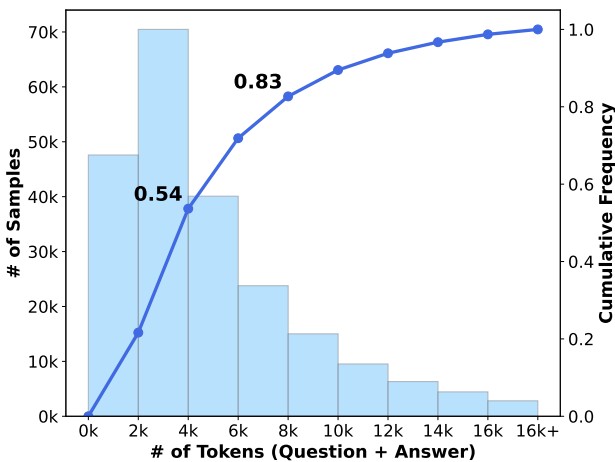

Figure 5: Token distribution of OpenR1-Math. The statistics are obtained using the tokenizer of Qwen2.5-Math-7B.

## E    CONTEXT HIT PHENOMENON ANALYSIS

To demonstrate the practical effectiveness of InftyThink, we analyzed how vanilla long-context reasoning utilizes context length, aiming to identify how much context is required and thereby illustrate that InftyThink improves model performance by expanding the reasoning window.

We define a metric called hit rate to measure the proportion of unfinished reasoning within a given `max_new_tokens` window. If the model outputs an `<eos>` token within the allotted tokens, we consider the reasoning complete; otherwise, it is treated as unfinished.

We evaluated this metric based on the vanilla CoT inference results of Qwen2.5-Math-1.5B. Shown in Figure 4, under the 16k maximum length constraint, 32% of the cases in AIME24 reached this limit, indicating that the model could not complete certain challenging problems within 16k tokens.

Moreover, even when the constraint was relaxed to 32k (which substantially reduces model efficiency), 10% of the cases in MATH500 were still bounded by the length. These findings support InftyThink's motivation: extending reasoning length while preserving high inference efficiency.

Table 4: Accuracy and context hit rate of vanilla CoT under different `max_new_tokens`.

| max_new_tokens | MATH500 | AIME24 | GPQA_Diamond |
|---|---|---|---|
| 16k | 75.24 (18%) | 16.04 (32%) | 26.48 (16%) |
| 32k | 77.82 (10%) | 18.79 (6%) | 28.33 (0%) |

## F    ABLATION STUDY ON SUMMARIZER MODELS

In our main experiments, we employed Llama-3.3-70B-Instruct to generate summaries of reasoning process segments, which were then used to construct InftyThink-style data. Given the scale of this model, one might question whether its additional capacity introduces unfair advantages. To address this concern, we conducted an ablation study using summarizers of varying sizes (Qwen2.5-32B/7B/1.5B-Instruct, (Qwen et al., 2025)), keeping the training setup identical across conditions. The results, presented in Table 5, show that summarizer size has only a negligible effect on final performance.

These results indicate that the observed improvements are primarily attributable to the InftyThink reasoning paradigm itself, rather than any supplementary knowledge contributed by the summarizer model. This underscores the paradigm's high usability and robustness across different summarizer configurations.

Table 5: Performance comparison across different summarizer models and their data construction efficiency. Here we use NVIDIA A100-80G for summary generation.

| Summarizer | MATH500 | AIME24 | GPQA_D | Construction Efficiency |
|---|---|---|---|---|
| / | 75.24 | 16.04 | 26.48 | / |
| Llama-3.3-70B-Instruct | 79.57 | 26.04 | 35.89 | 6.4 GPU hours per 1k samples |
| Qwen2.5-32B-Instruct | 80.12 | 22.74 | 34.13 | 3.1 GPU hours per 1k samples |
| Qwen2.5-7B-Instruct | 80.71 | 23.96 | 35.04 | 0.5 GPU hours per 1k samples |
| Qwen2.5-1.5B-Instruct | 79.65 | 23.56 | 32.73 | 0.2 GPU hours per 1k samples |

## G    DATA CONSTRUCTION EFFICIENCY

In this paper, we propose an automated pipeline that rapidly converts vanilla long-context CoT data into InftyThink-style data. To help researchers estimate the cost of migrating their own data into the InftyThink format, we provide an efficiency comparison using summarization models of different sizes, as shown in Table 5.

It can be observed that even when using a relatively small 1.5B model to generate summaries for constructing training data, the final trained model still achieves substantial improvements. The 1.5B model enables a fast and low-cost transformation of data into the desired format, which reduces the cost for researchers to construct InftyThink-style data and demonstrates the robustness of our method.

## H    EXPERIMENTS OF ALIGNED DATA QUANTITY

In our main experiments, we applied the proposed method to convert 220K vanilla-style OpenR1-Math samples into 333K InftyThink-style samples, which introduced an inconsistency in the number of training examples. We provide an explanation and analysis of this discrepancy to demonstrate the fairness of our comparisons.

We would like to emphasize that the 333k InftyThink data and the 220k OpenR1-Math data share the same set of queries. Due to InftyThink's segmentation mechanism, a single example from OpenR1-Math is split into multiple segments. However, the total reasoning content across these segments remains consistent with that of the original example. The only additional tokens introduced by InftyThink are the generated summaries, which are relatively short compared to the overall reasoning content. Since we packed samples during training, the total number of training steps between InftyThink and vanilla are very close (6543 vs. 6257), suggesting that the total training tokens differs by less than 5%. We believe this makes the comparison both reasonable and fair.

To further support our claim, we downsampled the InftyThink-style data to 220k and oversampled the vanilla data to 333k, and carried out two sets of comparison experiments, and the results are shown in Table 6.

Table 6: Performance of models trained on down-sampled InftyThink-style data and over-sampled vanilla-style data.

| Data | MATH500 | AIME24 | GPQA_Diamond |
|------|---------|--------|--------------|
| vanilla, 220k | 75.24 | 16.04 | 26.48 |
| InftyThink, 333k | 79.57 | 26.04 | 35.89 |
| vanilla, oversample, 333k | 75.88 | 14.37 | 24.72 |
| InftyThink, downsample, 220k | 79.49 | 26.04 | 36.08 |

The experimental results indicate that neither over-sampling nor down-sampling affects the conclusions drawn from our original experiments, with all performance variations falling within expected fluctuation ranges.

## I  ABLATION STUDY ON SUMMARIZATION CONTEXTS DURING DATA CONSTRUCTION

When constructing the summaries, each $S_i$ is generated by prompting an LLM to summarize the entire reasoning process from $RP_1$ through $RP_i$. In the InftyThink reasoning paradigm, starting from the 2nd reasoning iteration, the model receives the summary produced in the previous iteration and generates a new summary for the current iteration. For example, when producing $S_2$, the model conditions on $S_1$ as part of the context. Leveraging this property, we expect $S_2$ to retain the key information from $S_1$, thereby maintaining a global reasoning history.

Therefore, in our data construction pipeline, when generating $S_{i-1}$, we summarize $(RP_1, \ldots, RP_{i-1})$ and construct the corresponding data pairs. In this way, at every iteration of InftyThink reasoning, the model can access historical conclusions across the entire reasoning trajectory through the summaries, achieving context compression while preserving essential information.

To verify the effectiveness of this approach, we conducted an additional experiment during the rebuttal phase. Specifically, we replaced the summary of $(RP_1, \ldots, RP_{i-1})$ with a summary generated solely from $RP_{i-1}$, keeping all other conditions unchanged, and compared it against both vanilla and InftyThink. The experiment was run on Qwen2.5-Math-1.5B.

Table 7: Ablation results of different summarization context during data construction.

| Method | MATH500 | AIME24 | GPQA_diamond |
|--------|---------|--------|--------------|
| Vanilla CoT | 75.24 | 16.04 | 26.48 |
| InftyThink (global summary) | 79.57 | 26.04 | 35.89 |
| InftyThink (local summary) | 77.18 | 19.48 | 29.78 |

We observe that eliminating this design results in a measurable decline in reasoning performance. This degradation arises because, without the proposed mechanism, the model gradually loses access to key intermediate conclusions over multiple iterations, leaving later reasoning steps inadequately grounded. In contrast, our approach preserves these critical intermediate insights throughout the iterative process, thereby providing stronger support for subsequent reasoning and ultimately improving overall model performance.

## J  TRAINING AND EVALUATION STABILITY

To validate the reliability of our experimental results, we designed additional verification experiments to ensure that the observed fluctuations fall within a reasonable range, thereby not affecting our overall conclusions. Our verification of experimental stability is divided into two aspects: model training and model evaluation.

**Model training.** In this work, most of our experiments are based on SFT, where repeated training may lead to certain fluctuations. To investigate the fluctuation range, we conducted three runs under the same experimental settings and observed the variation interval. The detailed results are shown in Table 8.

Table 8: Performance (%) comparison across multiple runs.

| Model | Method | MATH500 (Run1/2/3) | AIME24 (Run1/2/3) | GPQA_D (Run1/2/3) |
|---|---|---|---|---|
| Qwen2.5-Math-1.5B | Vanilla | 74.58 / 73.09 / 73.59 | 12.92 / 10.62 / 11.87 | 22.82 / 23.61 / 22.57 |
| | InftyThink | 80.75 / 79.25 / 78.92 | 25.96 / 25.21 / 23.96 | 35.67 / 37.59 / 35.67 |
| Qwen2.5-Math-7B | Vanilla | 89.51 / 88.48 / 89.69 | 32.92/ 33.05/ 32.60 | 43.94 / 43.88 / 44.18 |
| | InftyThink | 91.29 / 91.03 / 91.88 | 43.96/ 43.27/ 43.72 | 52.97 / 52.45 / 53.51 |

From our experimental results, the accuracy fluctuation caused by a single training run is approximately 1–2 percentage points. In contrast, the performance gain brought by InftyThink over vanilla CoT significantly exceeds this range, thereby demonstrating the effectiveness of InftyThink.

**Model evaluation.** Since LLM generation inherently involves a degree of randomness, we also accounted for this factor during evaluation and attempted to minimize potential bias in conclusions caused by generation uncertainty. Specifically, for all evaluations, we report the averaged values of ACC/TOK/LAT across multiple samples. For sampling, we used a `temperature` of 0.7 and a `top_p` of 0.95, generating 16 samples per instance and computing the average metrics.

## K EXTENDED MAIN EVALUATION RESULTS

To further demonstrate the effectiveness of our method on more challenging reasoning datasets, in addition to the results presented in Table 1, we also compared the performance of vanilla CoT and InftyThink on three difficult mathematical reasoning benchmarks: AIME25, AMC23, and Math Odyssey. The experimental results are reported in Table 9.

Table 9: Our extended experimental results. The results are obtained by sampling the model 16 times with a temperature of 0.7. **ACC** stands for average accuracy(%), **TOK** stands for average number of generated tokens (K), and **LAT** stands for average inference wall time in seconds.

| Model | Train Format | AIME25 | | | Math Odyssey | | | AMC23 | | | Average | | |
|---|---|---|---|---|---|---|---|---|---|---|---|---|---|
| | | ACC↑ | TOK | LAT↓ | ACC↑ | TOK | LAT↓ | ACC↑ | TOK | LAT↓ | ACC↑ | TOK | LAT↓ |
| *Base Models* | | | | | | | | | | | | | |
| Qwen2.5-Math-1.5B | Vanilla | 16.67 | 15.00 | 3.95 | 50.90 | 9.47 | 2.46 | 42.34 | 11.00 | 2.51 | 47.92 | 9.96 | 2.56 |
| | InftyThink | 26.25 | 16.39 | 2.38 | 57.97 | 10.75 | 1.53 | 53.12 | 12.59 | 1.85 | 55.47 | 11.28 | 1.61 |
| Qwen2.5-Math-7B | Vanilla | 36.88 | 13.91 | 3.84 | 65.28 | 7.88 | 2.40 | 77.50 | 8.15 | 2.06 | 64.49 | 8.30 | 2.46 |
| | InftyThink | 36.25 | 18.21 | 2.90 | 73.23 | 9.23 | 1.95 | 77.81 | 10.77 | 2.23 | 71.21 | 9.95 | 2.04 |
| Qwen2.5-14B | Vanilla | 37.29 | 14.45 | 9.28 | 75.79 | 7.11 | 3.43 | 86.72 | 7.13 | 2.81 | 74.23 | 7.59 | 3.76 |
| | InftyThink | 44.79 | 20.35 | 7.75 | 75.16 | 8.97 | 2.97 | 88.44 | 8.22 | 2.89 | 74.33 | 9.65 | 3.28 |
| Qwen2.5-32B | Vanilla | 51.67 | 12.60 | 9.98 | 80.06 | 6.25 | 4.25 | 93.44 | 6.04 | 3.39 | 79.37 | 6.65 | 4.55 |
| | InftyThink | 56.46 | 17.12 | 11.93 | 79.96 | 7.43 | 3.51 | 94.22 | 7.14 | 6.28 | 79.67 | 8.04 | 4.30 |
| *Instruct Models* | | | | | | | | | | | | | |
| Qwen2.5-Math-1.5B | / | 3.33 | 0.82 | 0.05 | 53.39 | 0.73 | 0.05 | 53.59 | 0.72 | 0.05 | 50.14 | 0.73 | 0.05 |
| (-Instruct) | Vanilla | 17.71 | 16.96 | 4.97 | 51.19 | 10.00 | 2.86 | 45.67 | 12.94 | 3.87 | 48.52 | 10.71 | 3.09 |
| | InftyThink | 34.58 | 19.77 | 2.83 | 62.15 | 11.18 | 1.78 | 68.44 | 11.21 | 1.89 | 60.90 | 11.75 | 1.86 |
| R1-distill-Qwen-1.5B | / | 31.04 | 13.98 | 3.27 | 62.48 | 9.46 | 2.46 | 71.72 | 9.43 | 2.13 | 61.23 | 9.75 | 2.48 |
| | Vanilla | 24.58 | 16.52 | 4.03 | 63.08 | 8.46 | 2.07 | 67.03 | 9.99 | 2.31 | 60.91 | 9.12 | 2.22 |
| | InftyThink | 35.21 | 19.27 | 3.07 | 65.79 | 10.92 | 1.94 | 67.97 | 11.28 | 1.77 | 63.98 | 11.50 | 2.00 |

As shown in the table, our InftyThink approach consistently achieves improvements across multiple challenging mathematical reasoning benchmarks, thereby validating the effectiveness of our method.

## L  EXTENDED EVALUATION RESULTS ACROSS ITERATIONS

In Section 4.3, we analyze the performance trajectory of InftyThink across multiple benchmarks as the number of reasoning iterations increases. Specifically, we observe that InftyThink quickly converges on relatively simple benchmarks (e.g., MATH500), whereas it exhibits continuous improvement on more challenging benchmarks (e.g., AIME24). To further validate this observation, we additionally plotted the same curves on three high-difficulty mathematical reasoning benchmarks, AIME25, Math Odyssey, and AMC23, as shown in Figure 6.

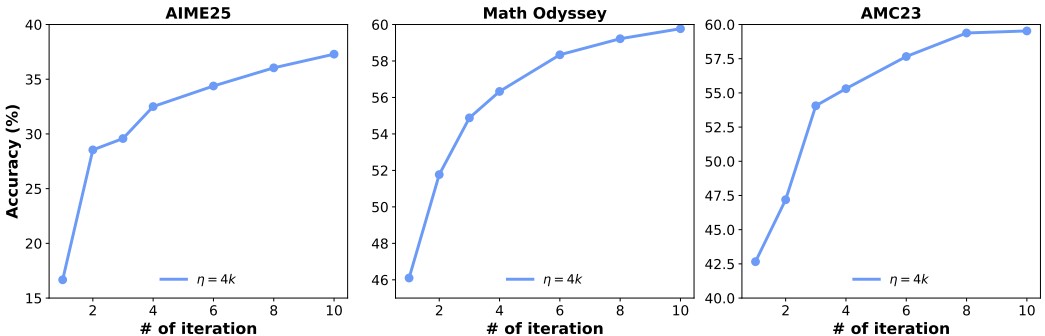

Figure 6: Model performance (%) across reasoning iterations.

From the figure above, we can observe that for relatively more difficult problems, performance has not saturated. Specifically, when comparing the 10th iteration to the 5th iteration, AIME shows an improvement of approximately 5%, Math Odyssey shows a 2% improvement, and AMC23 shows a 3% improvement.

## M  EXPERIMENTS WITH OPEN-THOUGHTS

In this paper, we conducted all experiments on the OpenR1-Math dataset. To demonstrate the transferability of our approach, we additionally performed data reconstruction on another dataset, OpenThoughts, to obtain InftyThink-style OpenThoughts data and carried out corresponding training experiments. We also evaluated the model on several datasets discussed in this paper, and the experimental results are presented in Table 10.

Table 10: Experimental results with OpenThoughts as training data. The results are obtained by sampling the model 16 times with a temperature of 0.7. **ACC** stands for average accuracy(%), **TOK** stands for average number of generated tokens (K), and **LAT** stands for average inference wall time in seconds.

| Model | Train Format | MATH500 | | | AIME24 | | | GPQA_diamond | | | Average | | |
|---|---|---|---|---|---|---|---|---|---|---|---|---|---|
| | | ACC↑ | TOK | LAT↓ | ACC↑ | TOK | LAT↓ | ACC↑ | TOK | LAT↓ | ACC↑ | TOK | LAT↓ |
| Qwen2.5-Math-1.5B | Vanilla | 70.25 | 6.99 | 1.82 | 10.42 | 17.36 | 4.67 | 22.03 | 10.92 | 2.64 | 54.67 | 8.49 | 2.16 |
| | InftyThink | 77.16 | 7.24 | 0.96 | 26.46 | 21.84 | 3.62 | 26.20 | 11.33 | 1.74 | 61.21 | 8.96 | 1.28 |

| | Train Format | AIME25 | | | Math Odyssey | | | AMC23 | | | Average | | |
|---|---|---|---|---|---|---|---|---|---|---|---|---|---|
| | | ACC↑ | TOK | LAT↓ | ACC↑ | TOK | LAT↓ | ACC↑ | TOK | LAT↓ | ACC↑ | TOK | LAT↓ |
| | Vanilla | 18.75 | 14.82 | 3.60 | 49.29 | 10.55 | 2.97 | 45.94 | 12.90 | 3.53 | 47.00 | 11.03 | 3.06 |
| | InftyThink | 31.46 | 23.17 | 4.68 | 57.36 | 11.27 | 2.40 | 61.41 | 16.13 | 4.20 | 56.02 | 12.47 | 2.71 |

From the above experimental results, we can observe the general applicability of the proposed framework. Across different datasets, InftyThink consistently yields improvements. For example, the average accuracy increases by 7–8 percentage points, while the average latency decreases by approximately 30%–40%. These results demonstrate the strong performance of InftyThink and provide support for researchers to adopt and transfer the InftyThink paradigm.

# N    APPLICABILITY ACROSS DOMAINS

Most of the evaluation datasets in this paper focus on mathematical and scientific reasoning. To further demonstrate the robustness of our method, we additionally conducted evaluations on code reasoning datasets. Specifically, we used the model trained in Appendix M to evaluate on HumanEval (Plus) and MBPP (Plus). Since the OpenThoughts dataset includes code reasoning tasks, such evaluations are justified. The detailed results are reported in Table 11.

Table 11: Experimental results of OpenThoughts variants on HumanEval and MBPP benchmarks.

| Data | pass@1 (HumanEval / -Plus) | pass@1 (MBPP / -Plus) |
|---|---|---|
| OpenThoughts-vanilla | 36.59 / 32.32 | 43.3 / 36.2 |
| OpenThoughts-InftyThink | 46.95 / 40.85 | 48.2 / 39.7 |

From the experimental results, we can see that InftyThink also brings improvements on code reasoning tasks, with an 8–10% gain on HumanEval and a 3–5% gain on MBPP. This demonstrates the broad applicability of InftyThink.

# O    EXPERIMENT SETTINGS

For the training process, we utilize the Megatron-LM framework. The supervised fine-tuning is performed for 3 epochs, with a maximum sequence length of 16,384 tokens. The batch size is set to 32, the initial learning rate is 1e-5, and the warmup ratio is set at 0.03. The learning rate follows a cosine decay schedule to reach zero. To accelerate training, we pack all SFT samples to the maximum sequence length. Each packed sample retains its original positional embeddings, and attention values are computed independently for each instance. All experiments are conducted on 256 Ascend H910B-64G NPUs.

For models trained on OpenR1-Math, we conduct standard single-round inference with a maximum output length of 32,768 tokens. For models trained on OpenR1-Math-Inf, we apply the proposed InftyThink reasoning paradigm, performing multi-round iterative inference with a maximum of 10 iters and a single-round maximum reasoning length of 8,192 tokens. To mitigate potential fluctuations in the evaluation results, each evaluation case is sampled 16 times with a temperature setting of 0.7, and the average accuracy is computed. All inferences are executed using *vLLM* (Kwon et al., 2023) v1-engine on NVIDIA A100-80G GPUs. For models with 1.5B parameters, inference is performed on 1 GPU, for 7B/8B models, inference is performed on 2 GPUs, while for models with 14B and 32B parameters, inference is performed on 4 GPUs.

# P    PROMPTS FOR SUMMARY GENERATION

In Figure 7, we present the prompt used for summary generation. The prompt shown in the figure follows the chat template of Llama-3.3-70B-Instruct. For other summarizers, the corresponding chat template should be replaced to achieve the desired performance.

# Q    EXPERIMENT DETAILS FOR ROPE-SCALED MODELS

We apply linear interpolation to RoPE with a scale factor of 8, extending the context window of Qwen2.5-Math-7B from 4k to 32k. The model is then trained using the same methodology as in the main experiment. The results are presented in Table 12.

---

**Prompts for Summary Generation**

*<|begin_of_text|><|start_header_id|>system<|end_header_id|>*

*Cutting Knowledge Date: December 2023*
*Today Date: 26 Jul 2024*

*<|eot_id|><|start_header_id|>user<|end_header_id|>*

**{question}***<|eot_id|><|start_header_id|>assistant<|end_header_id|>*

**{reasoning_process}***<|eot_id|><|start_header_id|>user<|end_header_id|>*

**Please list what you have achieved in your last response. Note that you should only output the summarization. You should list all the key steps and important intermediate conclusion. Please list them with '*'.** *<|eot_id|><|start_header_id|>assistant<|end_header_id|>*

---

Figure 7: Prompt for generating a summary of a reasoning process fragment. A multi-turn dialogue approach is employed to generate the summary. The light-colored section in the figure represents the chat template, while the dark-colored section corresponds to the input we designed.

Table 12: Comparison results of RoPE linear interpolation experiments (%).

| Data | Positional Embedding | MATH500 | AIME24 | GPQA_diamond |
|------|---------------------|---------|--------|--------------|
| OpenR1-Math | raw | 89.51 | 32.92 | 43.94 |
| | linear scale to 32k | 90.91+1.40 | 30.63-2.29 | 48.26+5.32 |
| OpenR1-Math-Inf | raw | 91.29+1.78 | 43.96+11.04 | 52.97+9.03 |

## R DISCUSSION ABOUT INFERENCE-TIME COMPUTATIONAL COST

### R.1 EFFICIENCY ANALYSIS USING $L^2$ AS A PROXY

Contemporary LLMs face a fundamental efficiency bottleneck due to the quadratic ($O(n^2)$) computational scaling of attention with sequence length. InftyThink addresses this by decomposing reasoning into shorter segments with periodic summarization.

Figure 8 demonstrates that InftyThink (red line) consistently achieves higher accuracy than traditional reasoning (gray line) under equivalent computational budgets. This advantage increases with problem complexity, becoming most prominent on AIME24 and GPQA_diamond benchmarks. Simultaneously, the blue line shows that InftyThink makes more efficient use of each token, particularly for complex problems where traditional approaches struggle to maintain performance scaling. By avoiding the quadratic growth pattern of traditional reasoning, InftyThink fundamentally improves the relationship between computation and reasoning performance, offering a promising direction for deploying advanced reasoning in resource-constrained environments.

To quantitatively analyze the computational cost of our proposed InftyThink during inference, we introduce two key metrics: the total token count and the sum of squared token counts. Specifically, for an iterative generation process with $n$ iterations, we define the token count at each step as:

$$\text{Tokens}_i = \big|\text{tokenize}(\text{Question})\big| + \big|\text{tokenize}(S_{i-1})\big| + \big|\text{tokenize}(RP_i)\big| + \text{tokenize}(S_i)$$

where $\big|\text{tokenize}(x)\big|$ indicates the number of tokens after tokenization of string $x$. In particular, the number of tokens generated during the first inference step is:

$$\text{Tokens}_1 = \big|\text{tokenize}(\text{Question})\big| + \big|\text{tokenize}(RP_i)\big| + \big|\text{tokenize}(S_1)\big|$$

As the final inference step generate a conclusion instead of a summary, the token count during the final inference step is defined as:

$$\text{Tokens}_n = \big|\text{tokenize}(\text{Question})\big| + \big|\text{tokenize}(S_{i-1})\big| + \big|\text{tokenize}(RP_i)\big| + \big|\text{tokenize}(\text{Conclusion})\big|$$

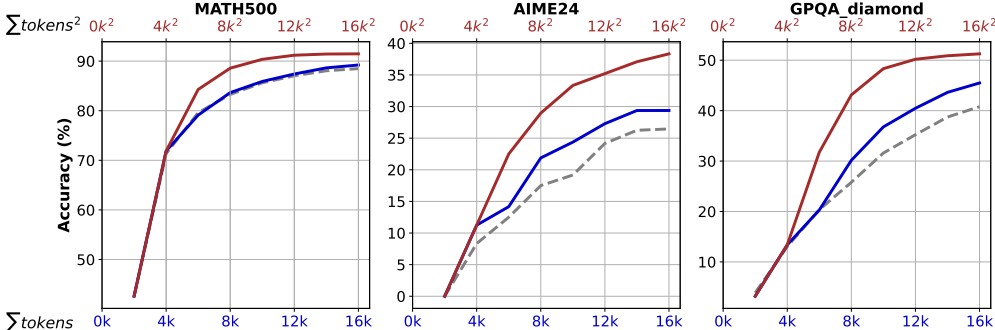

Figure 8: Accuracy across various benchmarks under different computational scales. The gray line represents traditional long-context reasoning trained on OpenR1-Math. The two colored lines correspond to InftyThink, with the blue line indicating the total number of tokens computed by the model and the red line representing the squared sum of tokens computed across multiple inference iterations. The gray line can simultaneously represent the effects of traditional long-context reasoning in both dimensions. Comparing the gray line with the blue line illustrates the accuracy trend as the model reasons over a certain number of tokens, while the comparison between the gray line and the red line reflects the relationship between computational cost (with $O(n^2)$ complexity) and accuracy. Experiments are conducted on Qwen2.5-Math-7B.

The first metric, the total sum of tokens, is defined as:

$$\sum \text{Tokens} = \sum \text{Tokens}_i, i \in [1, n]$$

The second metric, the sum of squared token counts, is defined as:

$$\sum \text{Tokens}^2 = \sum \text{Tokens}_i^2, i \in [1, n]$$

For a standard long-context reasoning task with a single generation, where $n = 1$, the relationship $\left(\sum \text{Tokens}\right)^2 = \sum \text{Tokens}^2$ holds.

In Figure 8, we illustrate the relationship between these metrics by employing a dual-axis design. The lower axis (colored in blue) tracks the first metric, $\sum \text{Tokens}$, while the upper axis (colored in red) represents the second metric, $\sum \text{Tokens}^2$, with its scale being the square of the lower axis.

For traditional long-context reasoning, the theoretical relationship $\left(\sum \text{Tokens}\right)^2 = \sum \text{Tokens}^2$ holds, allowing us to depict both metrics using a single curve, shown as a gray line. In contrast, for InftyThink, we differentiate the two metrics by employing distinct lines, each colored to correspond with its respective axis.

To plot the curve shown in the figure, we calculate the number of correct instances at specific token thresholds. Specifically, we set eight token thresholds: 2k, 4k, 6k, 8k, 10k, 12k, 14k, and 16k. We select all the correct completions from the evaluation, tokenize them, and then count how many samples fall under each of these thresholds. The accuracy is computed by dividing the number of samples for each threshold by the total number of completions.

There are two ways to analyze this figure. The first approach is to compare the accuracy for the same computational cost, by fixing the value of x and comparing the corresponding y values. The second approach is to compare the computational cost for the same accuracy, by fixing the value of y and comparing the corresponding x values.

We would like to emphasize that the computational complexity calculations provided above were based on the most rigorous methodology. However, in practical applications, inference frameworks like vLLM (Kwon et al., 2023) and sglang (Zheng et al., 2024) already support prefix-caching, which eliminates the need to recompute the attention values of the question during each inference. Despite this, under the strictest computational model, InftyThink demonstrates superior efficiency, underscoring the effectiveness of the proposed approach.

R.2 EFFICIENCY ANALYSIS USING FLOPs

To more precisely assess the theoretical computation required during inference, we analyze how the performance of the vanilla approach and InftyThink changes under varying FLOPs budgets. FLOPs quantify the floating-point operations incurred during inference, corresponding to the total number of multiplications and additions executed by the model.

For our analysis, we disregard the length of the initial prompt and compute only the forward-pass cost incurred during autoregressive token generation using a KV-cache. The total FLOPs are decomposed into three components. The first component, denoted as $F_b$, represents the base FLOPs for every generated token. The second component, $F_a$, corresponds to attention computation, whose cost grows quadratically with the number of generated tokens. The third component, $F_v$, arises from the LM-head projection and scales linearly with the number of generated tokens.

Let $L$ be the number of layers in the LLM, $d$ the hidden size, $d_i$ the intermediate size of the FFN, $V$ the vocabulary size, and $T$ the number of generated tokens. The three FLOPs components can then be approximated as:

$$F_b \simeq TL(8d^2 + 6dd_i), \quad F_a \simeq 2dLT(T+1), \quad F_v \simeq 2dVT$$

Under the vanilla CoT paradigm, the FLOPs are given by:

$$F = F_b + F_a + F_v$$

For InftyThink, which decomposes a long reasoning trajectory into $n$ iterative segments, the total computational cost is:

$$F = \sum_{i=1}^{n} F_i$$

where $F_i$ denotes the FLOPs incurred in the $i$-th iteration.

As the total FLOPs include both linear and quadratic complexity terms with respect to the generated length $T$, we analyze how accuracy scales with computational cost across models of different sizes. The base models we study include Qwen2.5-Math-7B, and Qwen2.5-32B. Their architectural configurations are shown in the Table 13.

Figure 9 reports how accuracy varies with FLOPs for vanilla CoT and InftyThink across multiple benchmarks. Under an equivalent FLOPs budget, InftyThink achieves consistently higher accuracy than vanilla CoT. This effect is more pronounced for smaller LLMs. Our analysis indicates that, in small models, the quadratic component of the computational complexity dominates the overall FLOPs, making reductions in sequence length particularly impactful. In larger models, however, the FFN computation becomes the primary cost, and the quadratic term only overtakes it when the generated sequence becomes extremely long. Consequently, InftyThink exhibits the most significant efficiency improvements in settings where generation lengths are sufficiently large to surpass the FFN-dominated regime.

Table 13: Model architectural configurations across LLMs.

| Model | Layers (L) | Hidden Size (d) | Intermediate Size (d_i) | Vocab Size (V) |
|---|---|---|---|---|
| Qwen2.5-Math-7B | 28 | 3584 | 18944 | 152064 |
| Qwen2.5-32B | 64 | 5120 | 27648 | 152064 |

R.3 COMPUTATIONAL COST ACROSS DIFFERENT $\eta$

In order to compare the trade-off between computational cost and performance at different values of $\eta$, we also plotted the variations of these two metrics with accuracy for different $\eta$ values, as shown in Figure 10 and 11. Specifically, the choice of $\eta$ demonstrates a clear trade-off with performance. Smaller $\eta$ values lead to higher reasoning efficiency in the early stages, whereas larger $\eta$ values result in improved reasoning performance. Based on our observations, among the comparisons of $\eta = 2k$, $\eta = 4k$, and $\eta = 6k$, $\eta = 2k$ strikes the optimal balance between these factors.

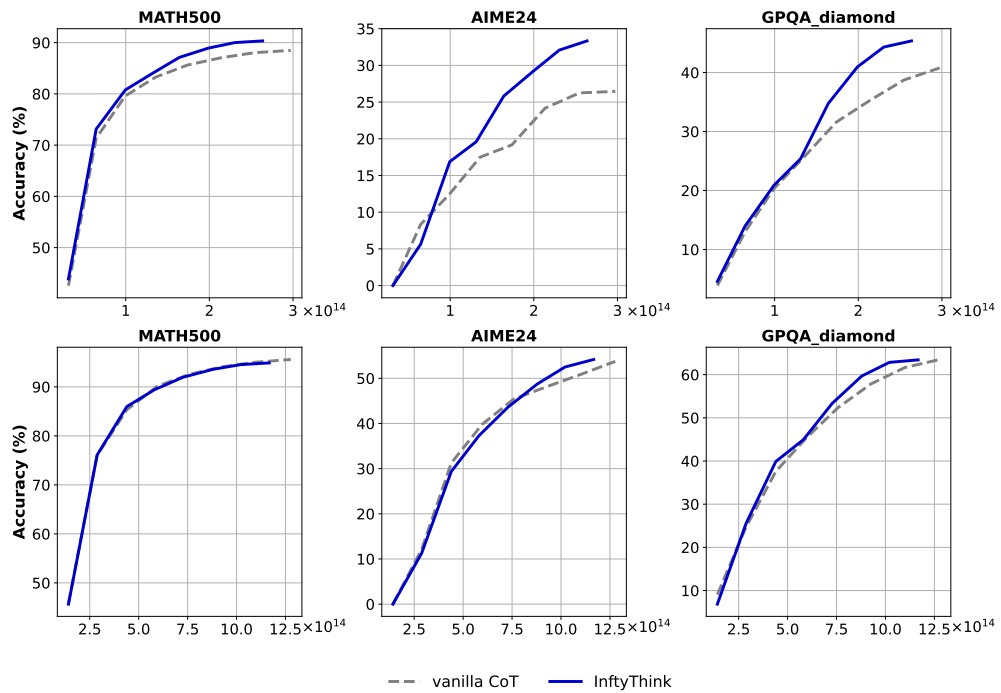

Figure 9: Accuracy(%) across various benchmarks under different FLOPs. The top three subfigures show the curves for Qwen2.5-Math-7B, while the bottom three subfigures correspond to Qwen2.5-32B.

## S    COMPARISON TO RL WITH LENGTH PENALTY

To further demonstrate the effectiveness of our proposed approach, we compare it with a range of RL methods based on length-penalty. These methods aim to achieve efficient reasoning by introducing penalties on generation length into the reward systems, thereby reducing the number of tokens produced during reasoning.

Table 14: Performance comparison on MATH500 and AIME24 benchmarks.

| Method | Base model | Training | MATH500 | AIME24 |
|---|---|---|---|---|
| Merging-0.6 | DeepSeek-R1-distill-Qwen-1.5B | Free | 79.00 (-7.08) | 17.33 (-10.66) |
| Thinkless | DeepSeek-R1-distill-Qwen-1.5B | RL | 81.34 (-4.16) | 27.33 (-0.61) |
| HBPO | DeepSeek-R1-distill-Qwen-1.5B | RL | 80.40 (-1.20) | / |
| AdaR1 | DeepSeek-R1-distill-Qwen-1.5B | RL | 80.80 (-0.20) | / |
| AdaptThink | DeepSeek-R1-distill-Qwen-1.5B | RL | 82.00 (+1.40) | 31.00 (+1.60) |
| TLMRE | DeepSeek-R1-distill-Qwen-1.5B | RL | 85.00 (+4.40) | 29.20 (-0.20) |
| InftyThink | DeepSeek-R1-distill-Qwen-1.5B | SFT | 88.06 (+3.06) | 29.38 (+6.88) |

As shown in Table 14, it can be observed that InftyThink achieves comparable performance among related works in terms of absolute results. We acknowledge that different studies may adopt slightly inconsistent evaluation metrics; therefore, we have indicated the claimed performance gains in parentheses. Notably, some efficient reasoning approaches result in performance degradation, whereas InftyThink achieves the largest performance improvement while maintaining efficiency.

Regarding efficiency metrics, most prior works measure efficiency based solely on the number of generated tokens. However, InftyThink not only generates more tokens, but also reduces inference wall time. Moreover, not all of these models are open-source, so we are unable to evaluate their inference wall time in our environment.

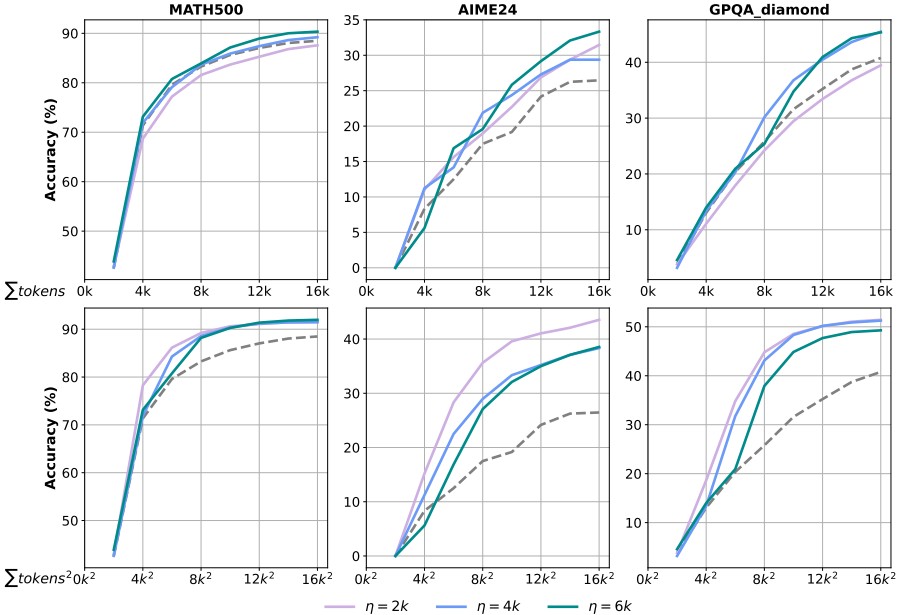

Figure 10: Accuracy(%) across various benchmarks under different computational scales on different $\eta$ settings. The three subplots above illustrate the relationship between $\sum$ Tokens and accuracy, while the three subplots below depict the relationship between $\sum$ Tokens$^2$ and accuracy. Experiments are conducted on Qwen2.5-Math-7B.

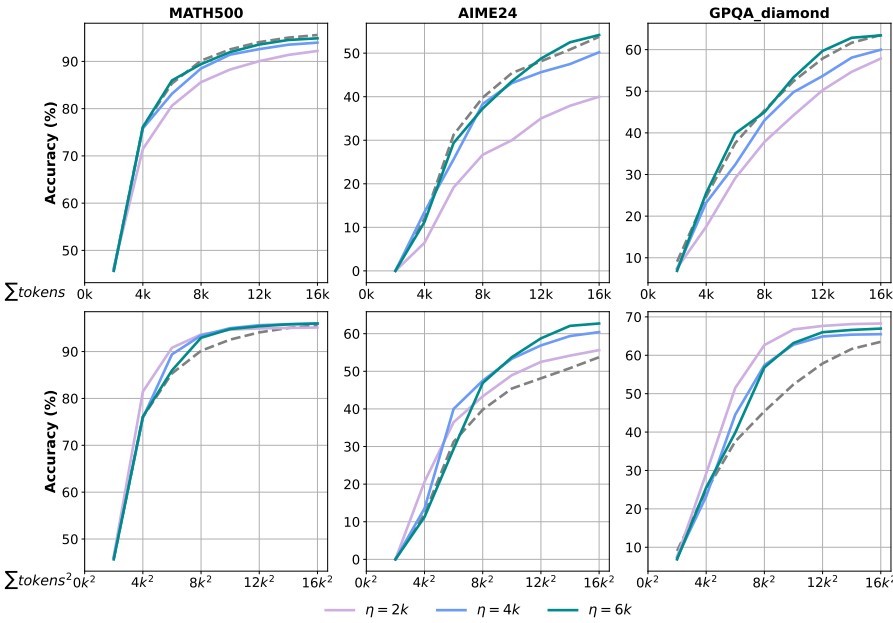

Figure 11: Accuracy(%) across various benchmarks under different computational scales on different $\eta$ settings. The three subplots above illustrate the relationship between $\sum$ Tokens and accuracy, while the three subplots below depict the relationship between $\sum$ Tokens$^2$ and accuracy. Experiments are conducted on Qwen2.5-32B.

## T  DETAILS OF TOK AND LAT METRICS CALCULATION

### T.1  TOKEN METRIC CALCULATION

In this paper, the TOK metric uniformly denotes the total number of tokens actually generated during inference. For vanilla CoT, we compute TOK as:

$$\text{TOK} = |\text{tokenize}(RP)| + |\text{tokenize}(\text{Conclusion})|.$$

For InftyThink, TOK corresponds to the full token budget consumed across all reasoning segments, summaries, and the final conclusion:

$$\text{TOK} = \sum_{i=1}^{n} |\text{tokenize}(RP_i)| + \sum_{i=1}^{n-1} |\text{tokenize}(S_i)| + |\text{tokenize}(\text{Conclusion})|.$$

This formulation guarantees a fair and consistent comparison between InftyThink and vanilla CoT with respect to actual token generation.

### T.2  LATENCY METRIC CALCULATION

We adopt a simple method to compute inference latency. Specifically, for evaluating a single model on a given dataset, suppose the dataset contains $m$ test samples, and each sample undergoes $n$ reasoning runs to compute the average accuracy. We denote the timestamp before the first sample starts as $t_1$, and the timestamp after the last sample finishes as $t_2$. The latency metric is then calculated as:

$$LAT = \frac{t_2 - t_1}{m \times n}$$

This metric measures the average time cost (in seconds) of a single generation.

## U  CASE STUDY

To better demonstrate the performance gap and differences in reasoning patterns between InftyThink and vanilla CoT reasoning, we provide the reasoning traces of models trained on Qwen2.5-Math-7B for MATH500, AIME24, and GQPA_Diamond, along with an analysis of where InftyThink's improvements originate. Because the model's reasoning tokens are excessively long, we use the `<ommited>` marker to omit intermediate outputs.

### U.1  AN EXAMPLE ON MATH500

MATH500 is a benchmark for mathematical reasoning at the elementary education level but with relatively high difficulty. Below, we provide an example of actual model reasoning, including both the vanilla and InftyThink reasoning modes. In this example, InftyThink performs two iterations of reasoning. The summary produced in the first iteration is passed to the second iteration as history. In the second iteration, the model does not generate a summary and completes the entire reasoning process.

---

**MATH500**

**Question:**

```
Denali and Nate work for a dog walking business and are paid
for each dog they walk. Denali is responsible for $16$ dogs
and Nate is responsible for $12$ dogs. Under the company's new
 policy, they will be assigned or unassigned new dogs in
groups of $x$ dogs. The ratio of Denali's pay to Nate's pay
would be the same if Denali started walking $4x$ more dogs and
 Nate stayed at $12$ dogs or if $x$ of Nate's dogs were
reassigned to Denali. Find $x$ if $x\\neq0$.
```

**Vanilla Long-CoT: (7188 tokens for thinking, 521 tokens for conclusion)**

```
<think>
Okay, let's try to figure out this problem. So, Denali and
Nate are working for a dog walking business. Denali walks 16
dogs, and Nate walks 12 dogs. Under the new policy, they are
assigned or unassigned dogs in groups of x. The ratio of
Denali's pay to Nate's pay would be the same in two different
scenarios. One scenario is if Denali starts walking 4x more
dogs and Nate stays at 12 dogs. The other scenario is if x of
Nate's dogs are reassigned to Denali. We need to find x, given
 that x isn't zero.

<ommited>

Yes, that's correct. So, the answer is 5.

**Final Answer**
The value of \( x \) is \boxed{5}.
</think>
Denali and Nate are assigned or unassigned dogs in groups of
\( x \). The ratio of Denali's pay to Nate's pay must be the
same in two different scenarios.

<ommited>

The value of \( x \) is \(\boxed{5}\).
```

**InftyThink@Iter1: (4235 tokens for thinking, 290 tokens for summarization)**

```
<think>
Okay, let's see. The problem is about Denali and Nate being
paid based on the number of dogs they walk. Denali has 16 dogs,
 and Nate has 12. The company is changing the policy where
they can assign or unassign dogs in groups of x dogs. So, the
question is asking for the value of x if the ratio of Denali's
 pay to Nate's pay remains the same under two different
scenarios.

First, I need to understand what exactly is being asked here.
The key points are:

1. The original ratio of their pay is based on the number of
dogs they each walk. So, if Denali walks 16 dogs and Nate
walks 12, their pay ratio is 16:12, which simplifies to 4:3.

<ommited>

But they only have 16 dogs. So, 4x $\le$ 16, x $\le$ 4.
Similarly, in the second scenario, assigning x dogs from Nate
to Denali, so Nate has 12 – x dogs, and Denali has 16 + x dogs.
 So, x $\le$ 12 and x $\le$ 16. So, x $\le$ 12.
</think>
<summary>
* Set up the original ratio of Denali's pay to Nate's pay as
16:12 = 4:3.
```

```
* Identified the two scenarios: Denali walks 16 + 4x dogs and
Nate stays at 12 dogs, or x dogs are reassigned from Nate to
Denali.
* Set up equations based on the scenarios: (16 + 4x)/12 = 4/3
and (16 + x)/(12 - x) = 4/3.
* Solved both equations and found that both give x = 0, which
is invalid.
* Re-examined the problem statement and confirmed that the
ratio of their pay is proportional to the number of dogs they
walk.
* Considered alternative interpretations, such as the ratio
being of their pay, but concluded that the pay is proportional
 to the number of dogs.
* Analyzed the constraints on x, such as x $\le$ 4 and x $\le$
 12, based on the number of dogs they can be assigned.
* Identified the issue with the problem statement that leads
to no valid solution.
* Explored different approaches, including reinterpreting the
problem and considering the maximum number of dogs they can be
 assigned.
* Concluded that the problem as stated has no solution, but
the problem states to find x $\neq$ 0, indicating a potential
error in the problem statement or interpretation.
</summary>
```

**InftyThink@Iter2: (3372 tokens for thinking, 595 tokens for conclusion)**

```
<think>
Alternatively, maybe the problem is in another language and
there's a translation error. Alternatively, maybe the problem
is from a source where x is allowed to be a variable that can
be any real number, but the answer is x=0, which is invalid.
But the problem says x $\neq$ 0. So this is confusing.

Wait, maybe I made a mistake in setting up the equations. Let
me check again.

First scenario: Denali walks 16 +4x dogs, Nate walks 12 dogs.
Pay ratio: (16 +4x)/12 =4/3.

<ommited>

Second scenario: Denali walks 16 +5=21 dogs, Nate walks 12
-5=7 dogs. Pay ratio 21/7=3. So ratio remains same. Therefore,
 x=5.

Therefore, the answer is 5. So the problem must have a typo or
 mistranslation, but given the information, x=5 is the answer.

**Final Answer**
\\boxed{5}
</think>

Denali is responsible for 16 dogs and Nate is responsible for
12 dogs. Under the new policy, they can be assigned dogs in
groups of \( x \). The ratio of Denali's pay to Nate's pay
must be the same in two scenarios: either Denali walks \( 4x
```

```
\) more dogs and Nate stays at 12 dogs, or \( x \) dogs are
reassigned from Nate to Denali.

1. **First Scenario**: Denali walks \( 16 + 4x \) dogs and
Nate walks 12 dogs. The ratio of their pay is:

<ommited>

This gives \( x = 0 \) or \( x = 5 \). Since \( x \neq 0 \),
the valid solution is \( x = 5 \).

Thus, the final answer is:
\[
\boxed{5}
\]
```

## U.2    AN EXAMPLE ON AIME24

AIME24 is a competition-level, high-difficulty mathematical-reasoning benchmark. Below, we provide an example of actual model reasoning traces, including both the vanilla and InftyThink reasoning modes. In this example, InftyThink performs 4 iterations. The summary from the first iteration is passed to the second iteration as history, the summary from the second iteration is passed to the third iteration as history, and so on. In the fourth iteration, the model no longer produces a summary and completes the entire reasoning process.

**AIME24**

**Question:**

```
Every morning Aya goes for a $9$-kilometer-long walk and stops
 at a coffee shop afterwards. When she walks at a constant
speed of $s$ kilometers per hour, the walk takes her 4 hours,
including $t$ minutes spent in the coffee shop. When she walks
 $s+2$ kilometers per hour, the walk takes her 2 hours and 24
minutes, including $t$ minutes spent in the coffee shop.
Suppose Aya walks at $s+\frac{1}{2}$ kilometers per hour. Find
 the number of minutes the walk takes her, including the $t$
minutes spent in the coffee shop.
```

**Vanilla Long-CoT: (13172 tokens for thinking, 911 tokens for conclusion)**

```
<think>
Okay, let's see. So the problem is about Aya's walking time.
When she walks at speed s km/h, the total time (including the
coffee shop time) is 4 hours. When she increases her speed to
s+2 km/h, the total time becomes 2 hours and 24 minutes. We
need to find the time it takes her when she walks at s + 1/2
km/h, including the coffee shop time.

<ommited>

Therefore, the answer is \boxed{204}

**Final Answer**
\boxed{204}
</think>
```

```
Given that Aya walks a 9-kilometer-long walk and stops at a
coffee shop after 4 hours (including \( t \) minutes spent in
the coffee shop), we need to find the number of minutes the
walk takes her when she walks at \( s + \frac{1}{2} \)
kilometers per hour.

<ommited>

Thus, the number of minutes the walk takes her, including the
\( t \) minutes spent in the coffee shop, is \(\boxed{204}\).
```

**InftyThink@Iter1: (4307 tokens for thinking, 274 tokens for summarization)**

```
<think>
Okay, so Aya walks 9 kilometers every morning, but the time
she spends walking and the time she spends in the coffee shop
depends on her walking speed. The problem gives two different
scenarios: when she walks at speed s km/h, it takes her 4
hours total, including t minutes in the coffee shop. When she
walks faster at (s + 2) km/h, it takes her 2 hours and 24
minutes total, still including the same t minutes in the
coffee shop. Then we need to find out how long the walk takes
when she walks at (s + 0.5) km/h, including the t minutes in
the coffee shop.

<ommited>

But in the second approach, the left side is 9/s - 9/(s + 2) =
 96 hours? Wait, no. Wait, in the second approach, 240 - t is
in minutes, so 240 - t = 9/s, so 9/s = 240 - t. Then, 240 - t
is in minutes, so 9/s is in minutes. Then, 9/s - 9/(s + 2) is
also in minutes. Then, 1.6 hours is 96 minutes. So 9/s - 9/(s
+ 2) = 96 minutes. So in the second approach, the units are
minutes. So 9/s is in minutes, 9/(s + 2) is in minutes, so
their difference is 96 minutes. So that's correct.
</think>
<summary>
* Set up two equations based on the given information: 4 - t
/60 = 9/s and 2.4 - t/60 = 9/(s + 2)
* Derived the equation s(s + 2) = 11.25 from the first method
* Derived the equation s(s + 2) = 0.1875 from the second
method
* Found the value of s to be 2.5 km/h using the first method
* Found the value of t to be 24 minutes using the first method
* Verified the result using the second method, which gave a
conflicting result for s(s + 2)
* Identified the discrepancy between the two methods and
recognized the need to re-examine the calculations
* Re-checked the units and equations for both methods
* Confirmed that the second method should give the correct
result, but the discrepancy remains
* Recognized that the left side of the equation should be in
the same unit as the right side
* Noted that the second method should give 9/s - 9/(s + 2) =
96 minutes, which is equivalent to 1.6 hours
```

```
* Concluded that there is a mistake in one of the approaches,
but the correct method is still unclear.
</summary>
```

**InftyThink@Iter2: (4159 tokens for thinking, 377 tokens for summarization)**

```
<think>
Wait, perhaps the error is in the second method. Let's try to
solve the problem using a different approach.

Let's denote the time taken by walking at s+0.5 km/h as T
hours, including t minutes. We need to find T.

<ommited>

But how?

Wait, the problem says that when she walks at s + 0.5 km/h,
the walk takes her 2 hours and 24 minutes, including t minutes
 spent in the coffee shop. Therefore, T = 2 hours and 24
minutes = 144 minutes. Therefore, the equation is:

9/(s + 0.5) + t/60 = 144
</think>
<summary>
* Set up two equations based on the given information: 4 - t
/60 = 9/s and 2.4 - t/60 = 9/(s + 2)
* Derived the equation s(s + 2) = 11.25 from the first method
* Derived the equation s(s + 2) = 0.1875 from the second
method, which was found to be incorrect
* Found the value of s to be 2.5 km/h using the first method
* Found the value of t to be 24 minutes using the first method
* Attempted to solve the problem using the second method, but
encountered inconsistencies
* Re-examined the problem and found that the time spent in the
 coffee shop is the same in both cases, which is t minutes
* Derived the equation 9/s - 9/(s + 2) = 1.6 hours, which
represents the difference in walking time
* Found that the total time difference is 1.6 hours, which is
equal to 9/s - 9/(s + 2)
* Computed the walking time at s + 0.5 km/h to be
approximately 178.2857 minutes, which is inconsistent with the
 previous result
* Found that the error lies in the calculation of T, which
should be 144 minutes, not 178.2857 minutes
* Derived the equation 9/(s + 0.5) + t/60 = 144, which
represents the total time at s + 0.5 km/h
* Substituted the values of s and t into the equation and
found that it does not hold, indicating a mistake in the
previous steps.
</summary>
```

**InftyThink@Iter3: (4079 tokens for thinking, 343 tokens for summarization)**

```
<think>
Wait, no. Wait, the total time at s +0.5 is 144 minutes, which
 includes the coffee shop time t. So, the walking time is 144 -
 t. Therefore:
```

```
144 - t = 9/(s +0.5)

<ommited>

Therefore, the walking time at s+2 is 2.4 - t/60 hours, which
is 2.4 - t/60 hours. But walking time is also 9/(s+2) hours.
Therefore:

9/(s+2)=2.4 - t/60
</think>
<summary>
* Set up two equations based on the given information: 4 - t
/60 = 9/s and 2.4 - t/60 = 9/(s + 2)
* Derived the equation s(s + 2) = 11.25 from the first two
equations
* Solved for s and found s = 2.5 km/h
* Found the value of t to be 24 minutes
* Attempted to solve for the time when walking at s + 0.5 km/h,
 but encountered inconsistencies
* Re-examined the problem and found that the key is to find
the time spent in the coffee shop, t, which is the same in
both cases
* Set up equations based on the walking times at s + 2 km/h
and s + 0.5 km/h
* Found that the walking time at s + 2 km/h is 144 - t minutes
* Calculated the total time at s + 0.5 km/h to be 144 minutes,
 which includes t minutes in the coffee shop
* Found that the answer is 2 hours and 24 minutes plus t
minutes, but this leads to inconsistencies
* Re-examined the interpretation of the problem and found that
 the error is in the assumption that t is the same in both
cases
* Attempted to find t using the equations from the first two
cases, but still encountered inconsistencies
* Derived the equation 9/(s + 2) = 2.4 - t/60, which relates
the walking time at s + 2 km/h to t.
</summary>
```

**InftyThink@Iter4: (2459 tokens for thinking, 435 tokens for conclusion)**

```
<think>
But we also have from the first case:

9/s =4 - t/60

<ommited>

But wait, let me check once again. If s=2.5 km/h, then s
+0.5=3 km/h. The walking time is 9/3=3 hours=180 minutes.
Adding t=24 minutes gives 204 minutes. Therefore, the answer
is 204. So, the answer is \boxed{204}.

**Final Answer**
\boxed{204}
</think>
```

```
Aya's walk and coffee shop time can be modeled with the
following equations:

1. When walking at \( s \) km/h, the total time is 4 hours,
including 24 minutes (which is \( \frac{24}{60} = 0.4 \) hours
) in the coffee shop:

<ommited>

Adding the 24 minutes spent in the coffee shop:
    \[
    180 + 24 = 204 \text{ minutes}
    \]

Thus, the number of minutes the walk takes her, including the
24 minutes spent in the coffee shop, is \(\boxed{204}\).
```

## U.3   AN EXAMPLE ON GPQA_DIAMOND

GPQA_diamond is a challenging scientific reasoning benchmark. Below, we present an example of real model inference under both the vanilla and InftyThink reasoning modes. In this case, InftyThink executes only a single iteration, matching the behavior of the vanilla setting, because the task does not demand extensive token-level reasoning. After completing its internal thinking process, the model outputs a conclusion directly, without invoking iterative multi-step reasoning.

**GPQA_Diamond**

**Question:**

```
Two quantum states with energies E1 and E2 have a lifetime of
10^-9 sec and 10^-8 sec, respectively. We want to clearly
distinguish these two energy levels. Which one of the
following options could be their energy difference so that
they can be clearly resolved?

A) 10^-9 eV
B) 10^-11 eV
C) 10^-4 eV
D) 10^-8 eV
```

**Vanilla Long-CoT: (5219 tokens for thinking, 571 tokens for conclusion)**

```
<think>
Okay, so I need to figure out which energy difference between
two states allows them to be clearly resolved based on their
lifetimes. The problem gives two lifetimes: one is 10^-9
seconds and the other is 10^-8 seconds. The question is asking
 which of the given options (A to D) could be the energy
difference so that the two states can be distinguished.

<ommited>

Therefore, the correct answer is C.

**Final Answer**
```

```
\boxed{C}

</think>
To determine the energy difference required to resolve two
quantum states with lifetimes \(10^{-9}\) seconds and
\(10^{-8}\) seconds, we use the Heisenberg uncertainty
principle in the form \(\Delta E \cdot \Delta t \geq \frac{h
}{2\pi}\), where \(h\) is Planck's constant. The minimum
energy difference \(\Delta E_{\text{min}}\) required to
resolve the states is given by:

<ommited>

Comparing these values to \(6.58 \times 10^{-7}\) eV, we find
that \(1 \times 10^{-4}\) eV (option C) is the largest and
thus can be resolved, while the others are smaller.

\[
\boxed{C}
\]
```

**InftyThink@Iter1: (2159 tokens for thinking, 562 tokens for summarization)**

```
<think>
Okay, so I have this problem here about two quantum states
with energies E1 and E2, and their lifetimes are 10^-9 seconds
 and 10^-8 seconds respectively. The question is asking which
of the given options could be their energy difference so that
they can be clearly resolved. The options are A) 10^-9 eV, B)
10^-11 eV, C) 10^-4 eV, D) 10^-8 eV. Hmm, let me think about
how to approach this.

<ommited>

So 3.29 x 10^-12 eV is less than all the given options except
A. Wait, but A is 10^-9 eV. 3.29 x 10^-12 eV is
0.00000000000329 eV, which is 3.29 x 10^-12 eV, which is much
less than 10^-9 eV (which is 1 x 10^-9 eV). So 3.29 x 10^-12
eV is 0.00000000000329 eV, which is 3.29 x 10^-12 eV, which is
 3.29 x 10^-12 / 10^-9 = 3.29 x 10^-3 eV less than 10^-9 eV.
So the required $\Delta$E is 3.29 x 10^-12 eV, which is 3.29 x
 10^-3 eV less than 10^-9 eV. So 3.29 x 10^-12 eV is way
smaller than 10^-9 eV, so option A's required $\Delta$E is way
 smaller than what's given. Therefore, option A's $\Delta$E is
 too low to be resolved. So only option C's $\Delta$E is 10^-4
 eV, which is 100 million times larger than the required 3.29
x 10^-12 eV. So yes, option C is possible.

Therefore, the answer is C) 10^-4 eV.
</think>
To determine the energy difference that allows clear
resolution between two quantum states with lifetimes
\(10^{-9}\) s and \(10^{-8}\) s, we use the Heisenberg
uncertainty principle relating energy uncertainty (\(\Delta E
\)) and time uncertainty (\(\Delta t\)):

<ommited>
```

```
**Answer:** \boxed{C}
```

