# OpenReview forum: "InftyThink: Breaking the Length Limits of Long-Context Reasoning in Large Language Models"
_ICLR.cc/2026/Conference — ICLR 2026 Poster_

### Official Review · Reviewer_xfMt · 2025-10-29

**Soundness:** 2
**Presentation:** 3
**Contribution:** 3
**Rating:** 2
**Confidence:** 4

**Summary:**

The paper proposes InftyThink, an iterative long-context reasoning method for LLMs. Instead of generating one long chain-of-thought, the model reasons in bounded segments of length η (e.g. 2K/4K/6K tokens), summarizes progress, and continues conditioning mostly on the summary. This is meant to (i) let models “think” for arbitrarily many steps, (ii) avoid quadratic attention over extremely long contexts, and (iii) improve reasoning accuracy at lower compute cost. The authors reconstruct a large supervised dataset (OpenR1-Math-Inf) by segmenting long math reasoning traces and adding summaries, and fine-tune various backbones (Qwen2.5-Math 1.5B/7B/etc., Llama-3.1-8B) on that data. They report higher accuracy on MATH500, AIME24, and GPQA_diamond, while claiming equal or lower latency compared to vanilla long-CoT decoding.

**Strengths:**

1. The core idea (periodic summarize-and-continue instead of one monolithic chain) is simple, intuitive, and architecturally lightweight — no special long-context transformer, just inference protocol + supervised finetuning.

2. The data pipeline that turns 220K long solutions into 333K iterative traces is clearly described and likely reusable for others training math/logic models.

3. Reported gains are consistent across multiple model sizes and families, especially on harder benchmarks like AIME24 and GPQA_diamond where deep multi-step reasoning matters.

**Weaknesses:**

1. Table 1 shows that in overall InftyThink emits more tokens per problem than vanilla (e.g. Qwen2.5-Math-1.5B on MATH500: 5.94K → 6.79K tokens), yet the reported latency drops (1.42s → 0.80s). This is confusing. How this translates to practical setup is unclear.

2. The InftyThink models depend critically on two hyperparameters: segment size η (they denote it like 2K/4K/6K) and max_iterations. Those exact values for each reported model are not given in Table 1. This makes it hard to reproduce or even interpret the latency numbers and token budget, since they obviously depend on η and how many iterations you unroll.

3. No variance / confidence intervals are provided in Table 1. For some benchmarks the absolute gain is only ~1–2%, which could easily fall within noise. The paper references “stability” in the appendix but does not surface it in the main results.

4. Figure 4 compares η = 2K / 4K / 6K. Converted to “total tokens generated,” the 2K model matches 4K and even beats 6K at similar total token counts. In other words, shorter segments with more iterations can be as good or better than longer segments, at roughly the same total text budget. This is surprising and important: it hints that extremely short steps plus good summaries may dominate, which would strongly shape how people deploy the method. The paper surfaces the curve, but gives almost no analysis of why 2K can keep up with or beat 4K/6K at equal effective cost. That missing discussion leaves open whether η is actually a tuning knob or just an implementation artifact.


I think that the pitch of longer reasoning with less latency and better accuracy, drop-in for any LLM is very strong, but the evidence for the latency/computation story is currently incomplete, and several presentation issues (Table 1 hyperparameters, missing variance, lack of explanation for Figure 4 behavior) are significant for an ICLR acceptance.

**Questions:**

1. How exactly were the latency numbers in Table 1 measured for vanilla vs. InftyThink? Were both sides run under identical inference infrastructure?

2. Why does InftyThink emit more total tokens but run faster? Please provide a step-by-step cost model and per-iteration timing.

3. What η and max_iterations were used for each model in Table 1? Were they tuned per task?

4. Please provide accuracy variance / CIs for at least the Table 1.

5. Figure 4. Why does η=2K match or outperform η=4K/6K at the same effective token budget? Is this simply because shorter segments force more frequent summarization (which the model has been trained to do well), or something else?

---

> ### Author Response · Authors · 2025-11-21
> **Rebuttal [1/4] - Briefly re-introducing InftyThink**
>
> Dear Reviewer xfMt,
>
> Thank you for the effort and time you dedicated to reviewing our paper, and for your recognition of the contribution of our proposed methods.
>
> ---
>
> Before addressing your concern, we would like to briefly revisit the motivation and working mechanism of InftyThink, and explain why InftyThink achieves both **higher performance** and **lower inference wall time** compared with vanilla CoT.
>
> The motivation behind InftyThink arises from the fact that modern reasoning models often generate extremely long chains-of-thoughts, typically exceeding 10K tokens or more. However, current decoder-based LLMs rely on self-attention, **whose computational complexity grows quadratically ($O(n^2)$) with the sequence length**. As a result, generating each additional token during late-stage reasoning incurs a rapidly increasing computational cost.
>
> To mitigate this $O(n^2)$ complexity, we introduce InftyThink, which **decomposes a long reasoning chain into multiple inference rounds, connected via summaries**. This design reduces the computational burden during inference. The relationship can be expressed as:
>
> $$
> R = R_1 + R_2 + \ldots + R_n
> $$
>
> $$
> R^2 \ge R_1^2 + R_2^2 + \ldots + R_n^2
> $$
>
> InftyThink achieves **better performance** mainly because:
>
> * It performs iterative reasoning, enabling reasoning over arbitrarily long contexts, which extends the reasoning context window.
> * It conducts reasoning within **shorter positional embedding ranges**, better leveraging the model’s pretraining context window.
>
> InftyThink achieves **higher efficiency** mainly because:
>
> * It **converts a single long generation into multiple short generations, greatly reducing the computational overhead induced by the decoder’s $O(n^2)$ complexity**. Consequently, the model maintains lower latency even when generating more total tokens. (See Figure 1 (a), the area under the curve.)
>
> ---

---

> ### Author Response · Authors · 2025-11-21
> **Rebuttal [2/4]**
>
> Below are our responses to the questions you pointed out:
>
> > **[Question 1]: Table 1 shows that in overall InftyThink emits more tokens per problem than vanilla (e.g. Qwen2.5-Math-1.5B on MATH500: 5.94K → 6.79K tokens), yet the reported latency drops (1.42s → 0.80s). This is confusing. How this translates to practical setup is unclear.
> How exactly were the latency numbers in Table 1 measured for vanilla vs. InftyThink? Were both sides run under identical inference infrastructure?**
>
> Thank you for your question. Regarding the calculation of latency, in the first version of our manuscript we briefly introduced the method in Appendix S. Specifically, **we compute the average latency by dividing the total inference time over the entire benchmark by the number of generated reasoning trajectories**. In the new revision, we further elaborate on the computation of the metrics (LAT and TOK) in more detail (in Appendix T of the new revision), making the experimental setup clearer.
>
> In addition, for all datasets, **both vanilla and InftyThink use exactly the same inference infrastructure, running on the same GPUs and under the same Docker environment**, ensuring fairness and the validity of the experimental results.
>
>
> > **[Question 2]: Why does InftyThink emit more total tokens but run faster? Please provide a step-by-step cost model and per-iteration timing.**
>
> Thank you for your question. As mentioned earlier, **InftyThink reduces the computational overhead arising from the $O(n^2)$ complexity of decoder-based models by decomposing a long reasoning process into multiple shorter inference steps**. This allows the model to generate more tokens while still lowering the overall inference latency.
>
> In addition, we have added a new Appendix section U illustrating the actual reasoning behaviors of both vanilla CoT and InftyThink on MATH500, AIME24, and GPQA_diamond, which helps readers better understand our method.
>
> > **[Question 3]: The InftyThink models depend critically on two hyperparameters: segment size η (they denote it like 2K/4K/6K) and max_iterations. Those exact values for each reported model are not given in Table 1. This makes it hard to reproduce or even interpret the latency numbers and token budget, since they obviously depend on η and how many iterations you unroll.
> What η and max_iterations were used for each model in Table 1? Were they tuned per task?**
>
>
> Thank you for your question. In the initial manuscript submission, we had already specified the values of these hyperparameters: $\eta$ is set to 4k (see line 293), and `max_iters` is fixed at 10 (see line 1184). To improve clarity and reproducibility, we will restate these key hyperparameters in Section 3.1. We appreciate you pointing out this ambiguity in the earlier description.
>
> Furthermore, unless explicitly stated in the ablation studies, both $\eta$ and `max_iters` remain fixed to these values across all models and all benchmarks used in this work.

---

> > ### Author Response · Authors · 2025-11-21
> > **Rebuttal [3/4] - Question 4**
> >
> > > **[Question 4]: No variance / confidence intervals are provided in Table 1. For some benchmarks the absolute gain is only ~1–2%, which could easily fall within noise. The paper references “stability” in the appendix but does not surface it in the main results.
> > Please provide accuracy variance / CIs for at least the Table 1.**
> >
> > Thank you for your question. First, during evaluation, we sample 16 responses and compute the average accuracy to reduce potential noise. This evaluation method is widely used in many other papers[1-2]. While training stability is indeed an important consideration, our method **requires a significant amount of computational resources (5250 A100 GPU hours on Qwen2.5-32B)**, making it difficult to conduct multiple training runs for noise analysis.
> >
> > We complete an additional experiment by randomly shuffuled data from both the vanilla and InftyThink datasets, and conducted 2 additional training runs on 1.5B model for each, aiming to assess the potential variance. This experimental result has already been included in Appendix I of the initial submission. During the rebuttal period, we conducted multiple runs on Qwen2.5-Math-7B, and the results also show that the variance range is smaller than the performance gain brought by InftyThink.
> >
> > |Base model|Method|MATH500 (Run1/Run2/Run3)|AIME24 (Run1/Run2/Run3)|GPQA_D (Run1/Run2/Run3)|
> > |-|-|-|-|-|
> > |Qwen2.5-Math-1.5B |Vanilla| 74.58 / 73.09 / 73.59| 12.92 / 10.62 / 11.87|22.82 / 23.61 / 22.57 |
> > | |InftyThink|80.75 / 79.25 / 78.92| 25.96 / 25.21/ 23.96|35.67 / 37.59 / 35.67|
> > |Qwen2.5-Math-7B |Vanilla| 89.51 / 88.48 / 89.69  | 32.92/ 33.05/ 32.60 | 43.94 / 43.88 / 44.18|
> > | |InftyThink| 91.29 / 91.03 / 91.88 | 43.96/ 43.27/ 43.72 | 52.97 / 52.45 / 53.51 |
> >
> > From our experimental results, the accuracy fluctuation caused by a single training run is approximately 1-2 percentage points. In contrast, **the performance gain brought by InftyThink over vanilla CoT significantly exceeds this range**, thereby demonstrating the effectiveness of InftyThink.
> >
> > Ref:
> > [1] DeepSeek-AI, DeepSeek-R1: Incentivizing Reasoning Capability in LLMs via Reinforcement Learning.
> > [2] Lou, et al. AdaCoT: Pareto-Optimal Adaptive Chain-of-Thought Triggering via Reinforcement Learning.

---

> ### Author Response · Authors · 2025-11-21
> **Rebuttal [4/4] - Question 5**
>
> > **[Question 5]: Figure 4 compares η = 2K / 4K / 6K. Converted to “total tokens generated,” the 2K model matches 4K and even beats 6K at similar total token counts. In other words, shorter segments with more iterations can be as good or better than longer segments, at roughly the same total text budget. This is surprising and important: it hints that extremely short steps plus good summaries may dominate, which would strongly shape how people deploy the method. The paper surfaces the curve, but gives almost no analysis of why 2K can keep up with or beat 4K/6K at equal effective cost. That missing discussion leaves open whether η is actually a tuning knob or just an implementation artifact.
> Figure 4. Why does η=2K match or outperform η=4K/6K at the same effective token budget? Is this simply because shorter segments force more frequent summarization (which the model has been trained to do well), or something else?**
>
> Thank you for raising this question. We address it from two complementary perspectives:
>
> * **On the behavior observed in the curves:** The phenomenon you mentioned, namely that “$\eta$ = 2K matches or exceeds the performance of $\eta$ = 4K/6K under the same effective token budget”, is **not a deterministic trend**. For instance, on MATH500, ($\eta$ = 2K, n = 2) performs worse than ($\eta$ = 4K, n = 1), and (η = 2K, n = 3) performs worse than ($\eta$ = 6K, n = 1). Similar patterns are observed on AIME24. Our analysis shows that **InftyThink is relatively insensitive to the choice of $\eta$**. When $\eta$ falls within a reasonable range (e.g., 2K–6K), the model exhibits comparable performance given the same token budget or sufficient reasoning iterations, consistent with the findings discussed in Section 4.2. Consequently, $\eta$ = 2K may occasionally outperform or underperform $\eta$ = 4K/6K. The reason $\eta$ = 2K eventually approaches the performance of $\eta$ = 4K/6K in the plotted results is that the curves use the **maximum number of reasoning iterations n**, which aggregates all successful samples with iteration counts $\le$ n. **As n increases, performance naturally converges to the model’s inherent reasoning capability**.
>
> * **On the practical selection of $\eta$:** We view the weak sensitivity of InftyThink to $\eta$ as an appealing property, as **it allows practitioners to tailor $\eta$ to different hardware constraints**. A larger $\eta$ reduces the number of summarization steps and better leverages existing GPU optimizations, while a smaller $\eta$ reduces the number of generated tokens per iteration, helping maintain a lower GPU memory footprint.
>
> We appreciate your feedback regarding the need for a more thorough analysis. In our revision, we expanded the discussions in Sections 4.2 and 4.3, as well as in the Appendix, to further strengthen the paper.
>
>
> ---
>
> Thank you again for the thorough review. We think that our rebuttal can address your main concern, if you have any further questions, we would be happy to answer them.
>
> Sincerely,
> InftyThink Authors

---

> ### Comment · Reviewer_xfMt · 2025-11-25
> **Questions are properly addressed.**
>
> I would like to thank authors for the detailed clarifications regarding my questions. I've adjusted my score.

---

> > ### Author Response · Authors · 2025-11-26
> >
> > We would like to once again thank you for the time and effort you devoted during the reviewing process and your kind re-consideration. Your feedback has helped us improve this paper, and we hope to have the opportunity for further discussions with you in the future.
> >
> > Thank you!

---

### Official Review · Reviewer_39tg · 2025-11-01

**Soundness:** 3
**Presentation:** 3
**Contribution:** 3
**Rating:** 8
**Confidence:** 3

**Summary:**

This paper introduces InftyThink, which is a paradigm for turning monolithic reasoning traces into a sequence of short reasoning segments interleaved with summaries of all previous reasoning segments, incrementally and iteratively generated. This idea of "summarize" and "reason" pairs can be used for reconstructing existing reasoning datasets in this iterative format.

The authors empirically investigate this approach for five base models of various sizes (1.5B to 32B parameters) they train.

For training they also reconstruct OpenR1-Math, using a larger model for the summarization (or comparable-sized ones: Appendix F) of the reasoning segments they partition the traces into.

They test their accuracy on MATH500, AIME 2024 and GPQA Diamond benchmarks (main text) and they report on performance benefits when compared to the vanilla models' case with performance gains most notable for smaller sizes. A discussion on the impact of eta (segment length) in evolving performance across summarizations further augments the analysis of the method.

**Strengths:**

- Excellent motivation and intuition.

- The sawtooth memory pattern offers a bounded memory footprint which is highly desired in applications.

**Weaknesses:**

- Requires a specialized data reconstruction pipeline for training, which in turn necessitates the use of an LLM (e.g. for summarization), so not a simple/lightweight process (even for the very small summarizers in Table 4 (Line 994)).

**Questions:**

- Partitioning the reasoning process is very important: on the one hand skewed partition sizes can lead to variable "compression" factors in generated summaries or skewed summaries (that could fail to fit into fixed context window sizes (eta)); on the other hand these imbalanced partitions could well be those that are semantically most reasonable. How do you plan to approach this? Fixed larger contexts? Smaller contexts? Variable contexts driven by semantics and heuristics for deciding semantic boundaries?

---

> ### Author Response · Authors · 2025-11-21
> **Rebuttal [1/1]**
>
> Dear Reviewer 39tg,
>
> Thank you for the effort and time you dedicated to reviewing our paper, and for your recognition of the contribution of our proposed methods. We have greatly benefited from your valuable feedback. Below are our responses to the weaknesses you pointed out:
>
> ---
>
> > **[Weakness 1]: Requires a specialized data reconstruction pipeline for training, which in turn necessitates the use of an LLM (e.g. for summarization), so not a simple/lightweight process (even for the very small summarizers in Table 4 (Line 994)).**
>
> Thank you for your question. Our proposed data construction pipeline does indeed rely on an LLM to generate summaries. However, as shown in Table 4, even a very small LLM is sufficient to produce high-quality summaries. **The cost of summary generation is relatively low** (approximately 2 A100-hours per 10K samples), which is almost negligible compared with distilling reasoning trajectories from powerful reasoning models such as DeepSeek-R1. Therefore, **we regard the data reconstruction method introduced in this paper as a post-processing approach** that directly improves both performance and efficiency through a change in the reasoning paradigm.
>
>
>
> > **[Question 1]: Partitioning the reasoning process is very important: on the one hand skewed partition sizes can lead to variable "compression" factors in generated summaries or skewed summaries (that could fail to fit into fixed context window sizes (eta)); on the other hand these imbalanced partitions could well be those that are semantically most reasonable. How do you plan to approach this? Fixed larger contexts? Smaller contexts? Variable contexts driven by semantics and heuristics for deciding semantic boundaries?**
>
> Thank you for your question. The parameter $\eta$ is a key component of InftyThink, as it defines the size of the context window used by the model when autonomously generating summaries. A larger $\eta$ allows more information to be preserved across iterations, whereas a smaller $\eta$ improves overall inference efficiency. In practice, the optimal choice of $\eta$ depends on the available hardware, latency requirements, and the context length of the base model used during SFT.
>
> In our work, we adopt **a relatively fixed $\eta$** to maintain stable GPU memory usage throughout the generation process and fully utilize computational resources. Importantly, our segmentation strategy is not strictly rigid: we enforce sentence-level integrity (as stated in lines 263–264), ensuring that no Reasoning Segment contains truncated sentences, and that the total number of tokens in each segment remains below the designated $\eta$.
>
> To understand whether a larger or smaller $\eta$ is preferable, we conduct an ablation study in Section 4.2. The results indicate that model performance is generally insensitive to $\eta$, but a value of 4k yields the strongest performance. We attribute this to the 4k pre-training context window of Qwen2.5-Math-7B: **aligning $\eta$ with the pre-training configuration more effectively activates the model’s learned knowledge and reasoning capabilities**. As discussed in Section 4.1, this alignment also explains why InftyThink can endow short-context LLMs with robust long-context reasoning capabilities.
>
> ---
>
> Thank you again for your thorough review. We hope that our rebuttal has addressed your concerns. If you have any further questions or suggestions, we would be more than happy to respond.
>
>
> Sincerely,
> InftyThink Authors

---

### Official Review · Reviewer_xrFD · 2025-11-01

**Soundness:** 3
**Presentation:** 3
**Contribution:** 3
**Rating:** 6
**Confidence:** 4

**Summary:**

The paper introduces an iterative summarization strategy that compresses earlier reasoning traces in real time. This strategy keeps the live context short, cuts computing costs, and, as demonstrated by evaluation on three math (MATH500, AIME24, and GPQA) and other benchmarks, improves task accuracy compared to vanilla CoT.

**Strengths:**

- The idea is very clear, well motivated, and easy to follow.
- The proposed approach allows models to “think” longer while maintaining a limited context size, reducing GPU RAM memory requirements.
- Informative appendix with many details, extra experiments, and analyses.

**Weaknesses:**

- There is no prompting-only pipeline ablation: it’s unclear how much of the gain comes from training on specific data rather than the summarize-and-continue procedure itself. How would your trained method (InftyThink) compare to a prompting-only pipeline on the same base model: run CoT, pause at fixed steps, prompt the model to summarize, feed the summary back, and continue without any extra fine-tuning. This would help isolate the value added by training and highlight the general impact of the proposed iterative reasoning approach.
- Source of metrics gains is unclear. Some baselines return no answer due to context limits (according to Appendix E, Qwen2.5-Math-1.5B hits context limit in 18% samples on MATH500 and AIME24 in 32% samples). It is not clear how much of the improvement comes from (a) solving previously unanswered cases vs. (b) better reasoning on cases vanilla CoT did answer incorrectly.
- The paper does not clearly show the break-even point where the method becomes faster/cheaper than vanilla CoT. The method adds generation of summaries, so the total number of tokens for InftyThink are roughly len(question) + len(summaries) + len(CoTs), which exceeds vanilla len(question) + len(CoTs). Where is the breaking point at which InftyThink becomes faster in latency / tokens per second than vanilla CoT?
- Using n^2 tokens as a proxy for inference cost can be misleading (Appendix Q). In practice, compute is often dominated by FFN layers at shorter contexts, with quadratic attention dominating only at larger contexts. A comparison in FLOPs or measured latency would be more convincing.

**Questions:**

Please address weaknesses as questions.

- Why is InftyThink's latency higher than that of vanilla CoT on AIME24 for Llama-3.1-8B and R1-distill-Qwen-7B?

---

> ### Author Response · Authors · 2025-11-21
> **Rebuttal [1/2]**
>
> Dear Reviewer xrFD,
>
> Thank you for reviewing our paper, acknowledging our motivation and methodology, and providing valuable feedback. Below, we address the weaknesses you have pointed out:
>
> ---
>
> > **[Weakness 1]: There is no prompting-only pipeline ablation: it’s unclear how much of the gain comes from training on specific data rather than the summarize-and-continue procedure itself. How would your trained method (InftyThink) compare to a prompting-only pipeline on the same base model: run CoT, pause at fixed steps, prompt the model to summarize, feed the summary back, and continue without any extra fine-tuning. This would help isolate the value added by training and highlight the general impact of the proposed iterative reasoning approach.**
>
> Thank you for raising this question. We consider it an important experiment and completed it during the rebuttal phase. **The results have been added as a new Section 4.4**, and we provide a brief summary of our findings here.
>
> Based on R1-distill-1.5B, we implemented InftyThink in a prompt-based manner. Specifically, our implementation follows exactly the approach you mentioned in your review. We used multiple models to diversify the summarizer, including the reasoner model itself (R1-distill-1.5B), a general-purpose model of the same size (Qwen2.5-1.5B-Instruct), and the model used for constructing our dataset (Llama-3.3-70B-Instruct).
>
> | **Reasoner model**              | **Summarizer model**        | **MATH500** | **AIME24** | **GPQA_D** |
> |--------------------------------|------------------------------|-------------|------------|------------|
> | R1-distill-Qwen-1.5B (tuned on OpenR1-Math)          | /                            | 85.54       | 24.79      | 34.69      |
> |          | InftyThink                   | 88.06       | 29.38      | 35.54      |
> | | R1-distill-Qwen-1.5B         | 83.63       | 26.85      | 26.76      |
> | | Qwen2.5-1.5B-Instruct        | 82.97       | 20.45      | 30.92      |
> |                                | Llama3.3-70B-Instruct        |      84.14 | 26.65  | 31.88   |
>
> The results show that implementing **InftyThink via a prompt-based strategy leads to a slight performance degradation**. We believe this is because the resulting reasoning paradigm differs from the one the model was trained on, preventing the model from fully exhibiting its optimal reasoning capability.
>
> In addition, we would like to highlight that in our implementation, the prompt-based approach noticeably increases the actual inference wall time, which diminishes InftyThink’s advantages in improving reasoning efficiency.
>
> We sincerely appreciate your question, which allowed us to further strengthen our paper.
>
> > **[Weakness 2]: Source of metrics gains is unclear. Some baselines return no answer due to context limits (according to Appendix E, Qwen2.5-Math-1.5B hits context limit in 18% samples on MATH500 and AIME24 in 32% samples). It is not clear how much of the improvement comes from (a) solving previously unanswered cases vs. (b) better reasoning on cases vanilla CoT did answer incorrectly.**
>
> Thank you for raising this question. During the rebuttal period, we added an additional experiment to disentangle the sources of improvement brought by InftyThink. Specifically, we relaxed the max_new_tokens limit of vanilla CoT to 32k and re-evaluated its performance.
>
> | max_new_tokens | MATH500 | AIME24 | GPQA_diamond |
> | --------       | -------- | -------- | --------  |
> |  16K           |  75.24   |  16.04   |   26.48   |
> |  32K           |  77.82   |  18.79   |   28.33   |
> |  InftyThink    |  79.57   |  26.04   |   35.89   |
>
> As shown in Appendix E, after increasing max_new_tokens to 32k, the context hit rate drops substantially (MATH500: 18% → 10%, AIME24: 32% → 6%, GPQA_diamond: 16% → 0%), yet the performance only improves marginally (around +2%). In contrast, InftyThink achieves up to ~10% improvement over vanilla CoT. This indicates that **InftyThink provides better reasoning rather than merely benefiting from longer generation length.**

---

> > ### Author Response · Authors · 2025-11-21
> > **Rebuttal [2/2]**
> >
> > > **[Weakness 3]: The paper does not clearly show the break-even point where the method becomes faster/cheaper than vanilla CoT. The method adds generation of summaries, so the total number of tokens for InftyThink are roughly len(question) + len(summaries) + len(CoTs), which exceeds vanilla len(question) + len(CoTs). Where is the breaking point at which InftyThink becomes faster in latency / tokens per second than vanilla CoT?**
> >
> > Thank you for raising this critical question, which is highly relevant to practical applications of the InftyThink reasoning paradigm. When writing the paper, we also considered conducting this analysis. However, we found it difficult to define a clear break-even point because InftyThink and vanilla CoT exhibit fundamentally different reasoning behaviors. Specifically, for the same problem, the two methods often generate CoT sequences of substantially different lengths, and this variation can differ widely across cases.
> >
> > In addition, we would like to highlight the proportion of summary tokens in our constructed training data. **These summaries account for only about 4% of all tokens.** Compared to the typically much longer CoT sequences, this proportion is extremely small, and we believe the associated computation time can be reasonably approximated as negligible.
> >
> >
> > > **[Weakness 4]: Using n^2 tokens as a proxy for inference cost can be misleading (Appendix Q). In practice, compute is often dominated by FFN layers at shorter contexts, with quadratic attention dominating only at larger contexts. A comparison in FLOPs or measured latency would be more convincing.**
> >
> > Thank you for pointing out this issue. We realized that the original calculation might not be sufficiently accurate. Therefore, **we have added a new analysis to evaluate the performance curve of InftyThink under different FLOPs budgets in Appendix R.2**. We greatly appreciate your suggestion, and we believe that a more accurate analysis will help strengthen our paper.
> >
> >
> > > **[Question 1]: Why is InftyThink's latency higher than that of vanilla CoT on AIME24 for Llama-3.1-8B and R1-distill-Qwen-7B?**
> >
> > Thank you for pointing out this issue. After examining the reasoning traces of both models and conducting multiple rounds of inference experiments, we attribute the observations to the following factors:
> >
> > + As detailed in Appendix S, we compute latency by dividing the total inference time across the entire benchmark by the total number of generated completions. This metric is sensitive to boundary cases, those whose reasoning trajectories require significantly more iterations than average. Since **all other completed cases must wait for these long-running instances to finish, the overall latency becomes inflated.** In our inspection of AIME24 reasoning traces, we found that, given the benchmark’s difficulty, the model typically performs about four iterations. However, roughly 6% of cases continue reasoning until they hit the predefined max_iters, thereby contributing disproportionately to waiting time. To address this, **we are updating the soon-to-be-released InftyThink inference script to adopt an asynchronous execution mode, allowing each case to complete independently**. This modification will more accurately reflect the efficiency gains of InftyThink.
> > + Moreover, due to the high difficulty of AIME24, InftyThink generates substantially more tokens than vanilla CoT across these models. We hypothesize that when the token count surpasses a certain proportion, determined by model architecture, scale, and optimization of the inference framework, InftyThink may incur higher latency. Nonetheless, this increased computation correlates with performance gains: we observe improvements of 14% on Llama-3.1-8B, 9% on Qwen2.5-Math-7B, and 5% on R1-distill-Qwen-1.5B (corrected from a typo in the previous version). Applying the asynchronous inference approach described in point 1 is likely to further improve overall efficiency.
> >
> > We sincerely appreciate your feedback; it has been instrumental in helping us refine our inference pipeline.
> >
> > ---
> >
> > Thank you again for your thorough review. We hope that our rebuttal has addressed your concerns. If you have any further questions or suggestions, we would be more than happy to respond.
> >
> > Sincerely,
> > InftyThink Authors

---

> > > ### Comment · Reviewer_xrFD · 2025-11-26
> > >
> > > Thanks for the comprehensive response and text revision. I've read it carefully and I'm updating scores accordingly.

---

> > > > ### Author Response · Authors · 2025-11-26
> > > >
> > > > We would like to once again thank you for the time and effort you devoted to the review process. Your review greatly helped us improve our paper, especially the newly added Section 4.4. We look forward to having the opportunity to discuss with you more deeply in the future.
> > > >
> > > > Thank you😄

---

### Official Review · Reviewer_wn6o · 2025-11-04

**Soundness:** 2
**Presentation:** 3
**Contribution:** 3
**Rating:** 6
**Confidence:** 4

**Summary:**

InftyThink proposes an iterative reasoning paradigm for large language models that breaks long, monolithic reasoning chains into multiple short segments with intermediate summarizations, enabling theoretically unbounded reasoning depth while keeping computational costs bounded. Experiments on different benchmarks and multi model architectures show that this approach improves reasoning accuracy and reduces inference latency, effectively overcoming context-length and efficiency limitations.

**Strengths:**

1. This paper proposes a novel paradigm that controls context length through iterative summarization, thereby overcoming the limitation of maximum context boundaries. The idea is both intuitive and reasonable, and the authors further construct corresponding datasets to enable models to acquire this capability.
2. Comprehensive experiments across different datasets and models demonstrate that this method improves model accuracy while reducing inference time, and it consistently proves effective across various model architectures.
3. The experiments on reasoning iteration rounds demonstrate that the proposed method can, to some extent, overcome the constraints of maximum context length.

**Weaknesses:**

1. During inference, the method shortens the context through summarization, which in turn reduces reasoning time. Although Table 1 provides evidence of this effect, the reduction in inference time is not particularly significant, and the paper lacks further analysis (e.g., detailed statistics showing the actual number of tokens used per reasoning step compared to the vanilla approach).
2. At each reasoning step, the model should only have access to \(S_{i-1}\). Why, when constructing the SFT data in Equation (2), is it based on \(RP_1, \ldots, RP_{i-1}\) rather than \(S_{i-1}\)?
3. In Figure 4, how are the different iterations obtained? My understanding is that the model generates multiple reasoning steps adaptively. Could you provide the inference prompt? In addition, the paper lacks some visualization analysis, such as a concrete example illustrating the model’s reasoning process.

**Questions:**

See the Weaknesses.

---

> ### Author Response · Authors · 2025-11-21
> **Rebuttal [1/2]**
>
> Dear Reviewer wn6o,
>
> Thank you for the effort and time you dedicated to reviewing our paper, and for your recognition of the contribution of our proposed methods. We have greatly benefited from your valuable feedback. Below are our responses to the weaknesses you pointed out:
>
> ---
>
> > **[Weakness 1]: During inference, the method shortens the context through summarization, which in turn reduces reasoning time. Although Table 1 provides evidence of this effect, the reduction in inference time is not particularly significant, and the paper lacks further analysis (e.g., detailed statistics showing the actual number of tokens used per reasoning step compared to the vanilla approach).**
>
> Thank you for your question. We address your concerns from two aspects: **efficiency gains** and **additional analysis**.
>
> + **Efficiency gains:** InftyThink is motivated by the quadratic complexity of sequence length in Transformer-based generation. By adopting an iterative-reasoning paradigm, it reduces the overall inference wall time while enabling the model to generate more tokens. As reported in Table 1, InftyThink decreases the inference latency of Qwen2.5-Math-1.5B by 23%–44%. On MATH500, it achieves a 44% reduction in inference time together with a 4.33% accuracy improvement, highlighting the effectiveness of the InftyThink reasoning framework.
> + **Additional analysis:** We provide extended analyses in both the Analysis section and the Appendix. For instance, Section 4.3 and Appendix K present performance trajectories across different InftyThink iterations. To better illustrate InftyThink’s behavior in practice, we added a new appendix section in the revised manuscript. This section compares vanilla CoT and InftyThink on MATH500, AIME24, and GPQA_diamond, and reports token usage for each component of every inference iteration, offering readers a more comprehensive understanding of InftyThink.
>
> > **[Weakness 2]: At each reasoning step, the model should only have access to (S_{i-1}). Why, when constructing the SFT data in Equation (2), is it based on (RP_1, \ldots, RP_{i-1}) rather than (S_{i-1})?**
>
> Thank you for raising this critical question. In the InftyThink reasoning paradigm, starting from the 2nd reasoning iteration, the model receives the summary produced in the previous iteration and generates a new summary for the current iteration. For example, when producing $S_2$, the model conditions on $S_1$ as part of the context. Leveraging this property, we expect $S_2$ to retain the key information from $S_1$, thereby **maintaining a global reasoning history**.
>
> Therefore, in our data construction pipeline, when generating $S_{i-1}$, we summarize $(RP_1, \ldots, RP_{i-1})$ and construct the corresponding data pairs. In this way, at every iteration of InftyThink reasoning, the model can access historical conclusions across the entire reasoning trajectory through the summaries, achieving context compression while preserving essential information.
>
> To verify the effectiveness of this approach, we conducted an additional experiment during the rebuttal phase. Specifically, we replaced the summary of $(RP_1, \ldots, RP_{i-1})$ with a summary generated solely from $RP_{i-1}$, keeping all other conditions unchanged, and compared it against both vanilla and InftyThink. The experiment was run on Qwen2.5-Math-1.5B.
>
>
> | Method | MATH500 | AIME24 | GPQA\_diamond |
> | -------- | -------- | -------- | -------- |
> | Vanilla CoT     | 75.24     | 16.04     |  26.48     |
> | InftyThink (global summary)    | 79.57     | 26.04     |  35.89     |
> | InftyThink (local summary) (new)    | 77.18     | 19.48     |  29.78     |
>
> We observe that eliminating this design results in a measurable decline in reasoning performance. This degradation arises because, without the proposed mechanism, the model gradually loses access to key intermediate conclusions over multiple iterations, leaving later reasoning steps inadequately grounded. In contrast, our approach **preserves these critical intermediate insights throughout the iterative process**, thereby providing stronger support for subsequent reasoning and ultimately improving overall model performance.

---

> ### Author Response · Authors · 2025-11-21
> **Rebuttal [2/2]**
>
> > **[Weakness 3]: In Figure 4, how are the different iterations obtained? My understanding is that the model generates multiple reasoning steps adaptively. Could you provide the inference prompt? In addition, the paper lacks some visualization analysis, such as a concrete example illustrating the model’s reasoning process.**
>
> Thank you for your question. We would also like to use this rebuttal to clarify a point that may have been insufficiently explained in the main text. In InftyThink’s reasoning framework, a key hyperparameter is the maximum number of reasoning iterations, **max\_iters**, which is set to 10 in this paper. After 10 iterations, if the model has not yet produced a final answer, we forcibly terminate the reasoning process. This plays a role analogous to the **max\_new\_tokens** constraint in vanilla CoT.
>
> Consequently, the results reported in Figure 4 are derived by evaluating performance under different values of **max_iters**. For instance, the accuracy reported at *iter = 4* denotes the proportion of questions that the model solves correctly **within the first four iterations**. This includes cases where the model requires only two or three iterations to reach the correct solution.
>
> In practice, for the sake of efficiency and to avoid redundant evaluations, we fix **max_iters = 10** during inference and record the model’s complete reasoning trajectory—including all *thinking traces*, *summaries*, and the final *conclusion* at each iteration. These stored trajectories allow us to compute, post hoc, the accuracy at any iteration cutoff, thereby enabling the construction of Figure 4.
>
> We have added this information in Appendix U to explicitly detail this reasoning procedure. We appreciate your valuable suggestion.
>
>
> ---
>
> Thank you again for the thorough review. We think that our rebuttal can address your main concern, if you have any further questions, we would be happy to answer them.
>
> Sincerely,
> InftyThink Authors

---

### Author Response · Authors · 2025-12-01
**Summary of Rebuttal & Discussion Phases**

Dear Reviewers, AC, SAC, and PC,

We sincerely appreciate the time and effort you devoted to reviewing our work, as well as the insightful feedback you provided, which has significantly strengthened our paper. Below, we summarize the progress so far.

---

We are encouraged to see that the reviewers recognized several key strengths of our work, including:

* **Well-motivated, simple-yet-effective paradigm:** This paper proposes an effective paradigm to address the limitations of reasoning models in both efficiency and context. (Reviewer `wn6o`, `xrFD`, `39tg`, `xfMt`)
* **Strong and consistent experimental results:** Through extensive experiments, InftyThink consistently and significantly improves both the performance and efficiency of reasoning models compared to the baseline. (Reviewer `wn6o`, `xfMt`)
* **Effectiveness in real-world applications:** The sawtooth memory pattern of InftyThink provides a bounded memory footprint, which is highly desirable in practical deployments. (Reviewer `xrFD`, `39tg`)
* **Comprehensive experiments and analyses:** The paper offers extensive implementation details, supplementary experiments, and analyses in both the main text and the appendix to thoroughly evaluate the proposed InftyThink reasoning paradigm. (Reviewer `wn6o`, `xrFD`)


---
During the rebuttal phase, we conducted additional experiments or supplemented relevant analyses in the revision to address the reviewers' concerns:

+ We added Appendix I, where we ablate the summary-based reasoning context used during data construction. The results show that **maintaining a globally persistent history leads to better performance.** (Reviewer `wn6o`)
+ In Section 4.4, we introduce a prompt-based baseline. Experimental results demonstrate that **our method consistently outperforms prompt-based approaches**, underscoring the necessity of the proposed training framework. (Reviewer `xrFD`)
+ In Appendix E, we provide a detailed decomposition analysis of InftyThink’s performance gains, indicating that **the improvements stem from enhanced reasoning capability, rather than merely from longer generated outputs.** (Reviewer `xrFD`)
+ In Appendix J, we add a set of experiments on training stability, further showing that **InftyThink’s improvements are well beyond the natural variance arising from training or evaluation fluctuations.** (Reviewer `xfMt`)
+ In Appendix R.2, we present **an analysis of the relationship between FLOPs and model performance**, offering a more rigorous demonstration of InftyThink’s practical efficiency gains. (Reviewer `xrFD`)
+ In Appendix T, we give **a more detailed explanation of the computation procedures for the TOK and LAT metrics**, reinforcing the soundness of our experimental methodology. (Reviewer `wn6o`, `xfMt`)
+ We additionally expand Appendix U with **real InftyThink reasoning examples**, providing a more intuitive illustration of how the proposed paradigm operates in practice. (Reviewer `wn6o`)

---
We also clarified several key points to address the reviewers' concerns:

+ We provide a more detailed exposition of InftyThink’s underlying working mechanism, explaining why it can simultaneously achieve higher accuracy while reducing inference latency. (Reviewer `xfMt`)
+ We further elaborate on the experimental results and observed phenomena, including the performance curves across iterations and the behavior of latency metrics. (Reviewer `wn6o`, `xrFD`)
+ We expand the discussion on the choice of the hyperparameter $\eta$, including how to select $\eta$ and comparisons across different values of $\eta$. (Reviewer `39tg`, `xfMt`)

---
We would like to briefly summarize the consensus reached with the reviewers during the discussion phase **before Nov 27th (the date of identities leaking)**. During the discussion, we are happy that:

+ After reading our re-introduction of InftyThink’s working mechanism, Reviewer `xfMt` gained a more accurate understanding of our method, considered their concerns properly addressed, and increased their rating from **2** to **6**. (26 Nov 2025)
+ After carefully reading our comprehensive response and the revised manuscript, Reviewer `xrFD` increased their rating from **6** to **8**. (26 Nov 2025)

In summary, before the incident, our ratings improved from **2/6/6/8** to **6/6/8/8**.

---
Once again, we sincerely thank all reviewers for their recognition and valuable suggestions. We believe that our responses address the raised concerns effectively, and we hope the program committee (reviewers, ACs, SACs, PCs) will take them into account in the next phases.

Sincerely,
InftyThink Authors

---

### Meta-Review · Area_Chair_y4P9 · 2025-12-06

**Summary:**

- The paper lacks further analysis of the reduction of inference time
- There is no prompting-only pipeline ablation
- Source of metrics gains is unclear
- The paper does not clearly show the break-even point where the method becomes faster/cheaper than vanilla CoT
- The InftyThink models depend critically on two hyperparameters
- No variance / confidence intervals are provided in Table 1
- The paper surfaces the curve, but gives almost no analysis of why 2K can keep up with or beat 4K/6K at equal effective cost

**Reviewer Concerns:**

All the reviewer concerns listed above were addressed by the rebuttal.

**Reviewer Scores:**

As the concerns are well addressed by the rebuttal, I guess:
- Reviewer xfMt will raise the score from 2 to 6.
- Reviewer xrFD will raise the score from 6 to 8.
The other two reviewers will keep their positive scores of 6 and 8.
So the final scores would be 6/6/8/8.

---

### Decision · Program_Chairs · 2026-01-26

Accept (Poster)